**A probabilistic framework for the cover effect in bedrock erosion**
Jens M. Turowski
*Helmholtzzentrum Potsdam, German Research Centre for Geosciences GFZ, Telegrafenberg, 14473*
*Potsdam, Germany, turowski@gfz-potsdam.de*
Rebecca Hodge
*Department of Geography, Durham University, Durham, DH1 3LE, United Kingdom,*
*rebecca.hodge@durham.ac.uk*
**Abstract**
The cover effect in fluvial bedrock erosion is a major control on bedrock channel morphology and long-
term channel dynamics. Here, we suggest a probabilistic framework for the description of the cover
effect that can be applied to field, laboratory and modelling data and thus allows the comparison of
results from different sources. The framework describes the formation of sediment cover as a function
of the probability of sediment being deposited on already alluviated areas of the bed. We define
benchmark cases and suggest physical interpretations of deviations from these benchmarks.
Furthermore, we develop a reach-scale model for sediment transfer in a bedrock channel and use it to
clarify the relations between the sediment mass residing on the bed, the exposed bedrock fraction and
the transport stage. We derive system time scales and investigate cover response to cyclic
perturbations. The model predicts that bedrock channels achieve grade in steady state by adjusting
bed cover. Thus, bedrock channels have at least two characteristic time scales of response. Over short
time scales, the degree of bed cover is adjusted such that the supplied sediment load can just be
transported, while over long time scales, channel morphology evolves such that the bedrock incision
rate matches the tectonic uplift or base level lowering rate.
**1.   Introduction**
Bedrock channels are shaped by erosion caused by countless impacts of the sediment particles they
carry along their bed (Beer and Turowski, 2015; Cook et al., 2013; Sklar and Dietrich, 2004). There are
feedbacks between the evolving channel morphology, the bedload transport, and the hydraulics
(e.g., Finnegan et al., 2007; Johnson and Whipple, 2007; Wohl and Ikeda, 1997). Impacting bedload
particles driven forward by the fluid forces erode and therefore shape the bedrock bed. In turn, the
morphology of the channel determines the pathways of both sediment and water, and the forces the
latter exerts on the former, and thus sets the stage for the entrainment and deposition of the
sediment (Hodge and Hoey, 2016). Sediment particles play a key role in this erosion process; they
provide the tools for erosion and also determine where bedrock is exposed such that it can be worn
away by impacting particles (Gilbert, 1877; Sklar and Dietrich, 2004).
The importance of the cover effect – that a stationary layer of gravel can shield the bedrock from
bedload impacts – has by now been firmly established in a number of field and laboratory studies
(e.g., Chatanantavet and Parker, 2008; Finnegan et al., 2007; Hobley et al., 2011; Johnson and
Whipple, 2007; Turowski and Rickenmann, 2009; Turowski et al., 2008; Yanites et al., 2011).
Sediment cover is generally modelled with generic relationships that predict the decrease of the
fraction of exposed bedrock area $A^*$ with the increase of the relative sediment supply $Q_s^*$, usually
defined as the ratio of sediment supply to transport capacity. Based on laboratory experiments and
simple modeling, Turowski and Bloem (2016) argued that the focus on covered area is generally
justified on the reach scale and that erosion of bedrock under a thin sediment cover can be
neglected. However, the behavior of sediment cover under flood conditions is currently unknown
and the assumption that the cover distribution at low flow is representative of that at high flow may
not be justified (cf. Beer et al., 2016; Turowski et al., 2008).
The most commonly used function to describe the cover effect is the linear decline (Sklar and
Dietrich, 1998), which is the simplest function connecting the steady state end members of an empty
bed when relative sediment supply $Q_s^* = 0$ and full cover when $Q_s^* = 1$:

$$A^* = \begin{cases} 1 - Q_s^* & \text{for} \quad Q_s^* < 1 \\ 0 & \text{otherwise} \end{cases}$$

(eq. 1)
In contrast, the exponential cover function arises under the assumption that particle deposition is
equally likely for each part of the bed, whether it is covered or not (Turowski et al., 2007).

$$A^* = \begin{cases} e^{-Q_s^*} & \text{for} \quad Q_s^* < 1 \\ 0 & \text{otherwise} \end{cases}$$

(eq. 2)
Here, $e$ is the base of the natural logarithm.
Hodge and Hoey (2012) obtained both the linear and the exponential functions using a cellular
automaton (CA) model that modulated grain entrainment probabilities by the number of
neighbouring grains. However, consistent with laboratory flume data, the same model also produced
other behaviours under different parameterisations. One alternative behavior is runaway alluviation,
which was attributed by Chatanantavet and Parker (2008) to the differing roughness of bedrock and
alluvial patches. Due to a decrease in flow velocity, an increase in surface roughness and differing
grain geometry, the likelihood of deposition is higher over bed sections covered by alluvium
compared to smooth, bare bedrock sections (Hodge et al., 2011). This can lead to rapid alluviation of
the entire bed once a minimum fraction has been covered. The relationship between sediment flux
and cover is also affected by the bedrock morphology; flume experiments have demonstrated that
on a non-planar bed the location of sediment cover is driven by bed topography and hydraulics (e.g.,
Finnegan et al., 2007; Inoue et al., 2014). Johnson and Whipple (2007) observed that stable patches
of alluvium tend to form in topographic lows such as pot holes and at the bottom of slot canyons,
whereas Hodge and Hoey (2016) found that local flow velocity also controls sediment cover location.
The relationship between roughness, bed cover and incision was explored in a number of recent
numerical modeling studies. Nelson and Seminara (2011, 2012) were one of the first to model the
impact that the differing roughness of bedrock and alluvial areas has on sediment patch stability.
Zhang et al. (2014) formulated a macro-roughness cover model, in which sediment cover is related to
the ratio of sediment thickness to bedrock macro-roughness. Aubert et al. (2016) directly simulated
the dynamics of particles in a turbulent flow and obtained both linear and exponential cover
functions. Johnson (2014) linked sediment transport and cover to bed roughness in a reach-scale
model. Using a model formulation similar to that of Nelson and Seminara (2011), Inoue et al. (2016)
reproduced bar formation and sediment dynamics in bedrock channels. All of these studies used
slightly different approaches and mathematical formulations to describe alluvial cover, making a
direct comparison difficult.
Over time scales including multiple floods, the variability in sediment supply is also important (e.g.,
Turowski et al., 2013). Lague (2010) used a model formulation in which cover was written as a
function of the average sediment depth to upscale daily incision processes to long time scales. He
found that over the long term, cover dynamics are largely independent of the precise formulation at
the process scale and are rather controlled by the magnitude-frequency distribution of discharge and
sediment supply. Using the CA model of Hodge and Hoey (2012), Hodge (in press) found that, when
sediment supply was very variable (alternating large pulses with no sediment supply), the amount of
sediment cover was primarily determined by the recent supply history, rather than by the
relationships identified under constant sediment supply.

So far, it has been somewhat difficult to compare and discuss the different cover functions obtained
from theoretical considerations, numerical models, and experiments, since a unifying framework and
clear benchmark cases have been missing. Here, we propose such a framework, and develop type
cases linked to physical considerations of the flow hydraulics and sediment erosion and deposition.
We show how this framework can be applied to data from a published model (Hodge and Hoey,
2012). Furthermore, we develop a reach-scale erosion-deposition model that allows the dynamic
modeling of cover and prediction of steady states. Thus, we clarify the relationship between cover,
deposited mass and relative sediment supply. As part of this model framework we investigate the
response time of a channel to a change in sediment input, which we illustrate using data from a
natural channel.

## 2. A probabilistic framework

### 2.1. Development
Here we build on the arguments put forward by Turowski et al. (2007) and Turowski (2009). Consider
a bedrock bed on which sediment particles are distributed. We can view the deposition of each
particle as a random process, and each area element on the bed surface can be assigned a probability
for the deposition of a particle. When assuming that a given number of particles are distributed on
the bed, the mean behavior of the exposed area $A^*$ can be calculated from the following equation
(Fig. 1):
$$dA^* = -P(A^*, M_s^*, \dots)dM_s^*$$

(eq. 3)
$P$ is the probability that a given particle is deposited on the exposed part of the bed, which here is a
function of the fraction of exposed area ($A^*$) and a dimensionless mass of particles on the bed per
area ($M_s^*$, explained below), but which can be expected to also be a function of the relative sediment
supply, the bed topography and roughness, the particle size, the local hydraulics or other control
variables. $M_s^*$ is a dimensionless mass equal to the total mass of the particles residing on the bed per
area, which is suitably normalized. A suitable mass for normalization is the minimum mass required
to cover a unit area, $M_0$, as will become clear later. The minus sign is introduced because the fraction
of the exposed area reduces as $M_s^*$ increases. As most previous relationships are expressed in terms
of relative sediment supply $Q_s^*$, the relation of $M_s^*$ to $Q_s^*$ will be discussed later.

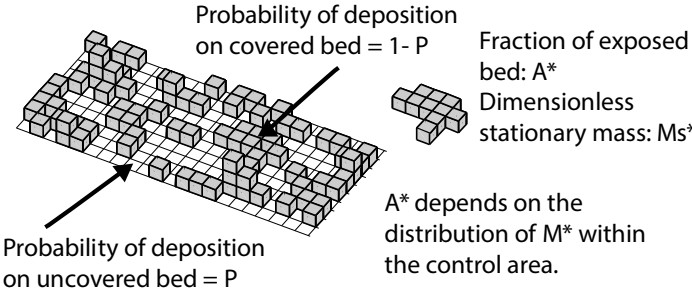


Fig. 1: Cartoon illustration of a bed partially covered by sediment. For purpose of illustration, the bed
is divided into a square raster, with each pixel of the size of a single grain. For a given number of
particles in the area of the bed of interest, the exposed area fraction of the bed is dependent on the
distribution of particles. Grains that sit on top of other grains do not contribute to cover. The
probability that a new grain is deposited on uncovered bed is given by $P$ (eq. 3).

We can make some general statements about $P$. First, $P$ is defined for the range $0 \leq A^* \leq 1$ and
undefined elsewhere. Second, $P$ takes values between zero and one for $0 \leq A^* \leq 1$. Third, $P(A^*=0) = 0$
and $P(A^*=1) = 1$. Note that $P$ is not a distribution function and therefore does not need to integrate
to one. Neither does it have to be continuous and differentiable everywhere.

For purpose of illustration, we will next discuss two simple forms of the probability function $P$ that
lead to the linear and exponential forms of the cover effect, respectively. First, consider the case that
all particles are always deposited on exposed bedrock. In this case, formally, to keep with the
conditions stated above, we define $P = 1$ for $0 < A^* \leq 1$ and $P = 0$ for $A^* = 0$. Thus, we can write

$$dA^* = -dM_s^* \quad \text{for} \quad 0 < A^* \leq 1$$
$$dA^* = 0 \qquad \text{for} \quad A^* = 0$$

(eq. 4)
Integrating, we obtain:
$$A^* = -M_s^* + C$$

(eq. 5)
where the constant of integration $C$ is found to equal one by using the condition $A^*(M_s^*=0) = 1$. Thus,
we obtain a linear cover function. Note that the linear cover function gives a theoretical lower bound
for the amount of cover: it arises when all available sediment always falls on uncovered ground, and
thus no additional sediment is available that could facilitate quicker alluviation. In essence, this is a
mass conservation argument. Now it is obvious why $M_0$ is a convenient way to normalize: in plots of
$A^*$ against $M_s^*$, we obtain a triangular region bounded by the points [0,1], [0,0] and [1,0] in which the
cover function cannot exist (Fig. 2).

Similarly to above, if we set $P$ to a constant value, $k$, smaller than one for $0 < A^* \leq 1$, we obtain

$$A^* = 1 - kM_s^*$$

(eq. 6)
It is clear that the assumption of $P = k$ is physically unrealistic, because it implies that the probability
of deposition on exposed ground is independent of the amount of uncovered bedrock. Especially
when $A^*$ is close to zero, it seems unlikely that, say, always 90% of the sediment falls on uncovered
ground. A more realistic assumption is that the probability of deposition on uncovered ground is
independent of location and other possible controls, but is equal to the fraction of exposed area, i.e.,
$P = A^*$. In a probabilistic sense, this is also the simplest plausible assumption one can make. Then

$$dA^* = -A^* dM_s^*$$

(eq. 7)
giving upon integration
$$A^* = e^{-M_s^*}$$

(eq. 8)
The argument used here to obtain the exponential cover effect in eq. (8) essentially corresponds to
the one given by Turowski et al. (2007). Since this case presents the simplest plausible assumption,
we will use it as a benchmark case, to which we will compare other possible functional forms of $P$.

In principle, the probability function $P$ can be varied to account for various processes that make
deposition more likely either on already covered ground by decreasing $P$ for the appropriate range of
$A^*$ from the benchmark case $P = A^*$, or on uncovered ground by increasing $P$ from the benchmark
case $P = A^*$. As has been identified previously (Chatanantavet and Parker, 2008; Hodge and Hoey
2012), roughness feedbacks to the flow can cause either case depending on whether subsequent
deposition is adjacent to or on top of existing sediment patches. In the former case, particles residing
on an otherwise bare bedrock bed act as obstacles for moving particles, and create a low-velocity
wake zone in the downstream direction. In addition, particles residing on other single particles are
unstable and stacks of particles are unlikely. Hence, newly arriving particles tend to deposit either
upstream or downstream of stationary particles and the probability is generally higher for deposition
on uncovered ground than in the benchmark case. In the latter case, larger patches of stationary
particles increase the surface roughness of the bed, thus decreasing the local flow velocity and
stresses, making deposition on the patch more likely. In this way, the probability of deposition on
already covered bed is increased in comparison to the benchmark case.

A simple functional form that can be used to take into account either one of these two effects is a
power law dependence of $P$ on $A^*$, taking the form $P = A^{*\alpha}$ (Fig. 2A). Then, the cover function
becomes (Fig. 2B):

$$A^* = (1 - (1 - \alpha)M_s^*)^{\frac{1}{1-\alpha}}$$
(eq. 9)
Here, the probability of deposition on uncovered ground is increased in comparison to the
benchmark exponential case if $0 < \alpha < 1$, and decreased if $\alpha > 1$.

A convenient and flexible way to parameterize $P(A^*)$ in general is the cumulative version of the Beta
distribution, given by:
$$P(A^*) = B(A^*; a, b)$$
(eq. 10)
Here, $B(A^*;a,b)$ is the regularized incomplete Beta function with two shape parameters $a$ and $b$,
which are both real positive numbers, defined by:
$$B(A^*; a, b) = \frac{\int_0^{A^*} y^{a-1}(1 - y)^{b-1}dy}{\int_0^1 y^{a-1}(1 - y)^{b-1}dy}$$
(eq. 11)
Here, $y$ is a dummy variable. With suitable choices for $a$ and $b$, cover functions resembling the
exponential ($a=b=1$), the linear form ($a=0$, $b>0$), and the power law form ($a>>b$ or $a<<b$) can be
retrieved. Wavy functions are also a possibility (Fig. 3), thus both of the roughness effects described
above can be modelled in a single scenario. Unfortunately, the integral necessary to obtain $A^*(M_s^*)$
does not give a closed-form analytical solution and needs to be computed numerically.

In principle, a suitable function $P$ could also be defined to account for the influence of bed
topography on sediment deposition. Such a function is likely dependent on the details of the
particular bed, hydraulics and sediment flow paths in a complex way and needs to be mapped out
experimentally.

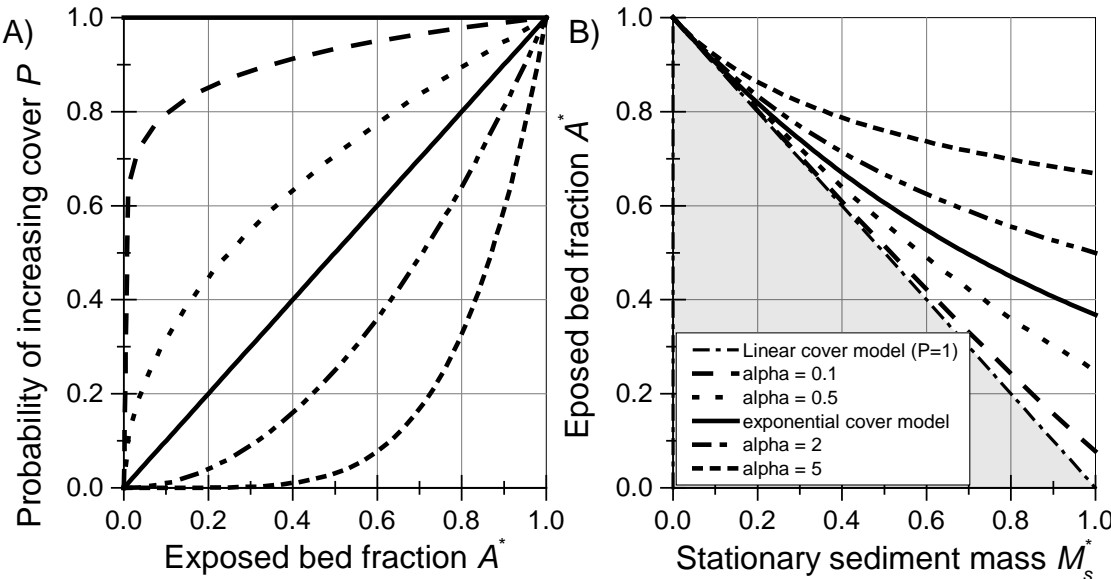


Fig. 2: A) Various examples for the probability function $P$ as a function of bedrock exposure $A^*$. B)
Corresponding analytical solutions for the cover function between $A^*$ and dimensionless sediment
mass $M_s^*$ using eq. (6), (7) and (9). Grey shading depicts the area where the cover function cannot
run due to conservation of mass.

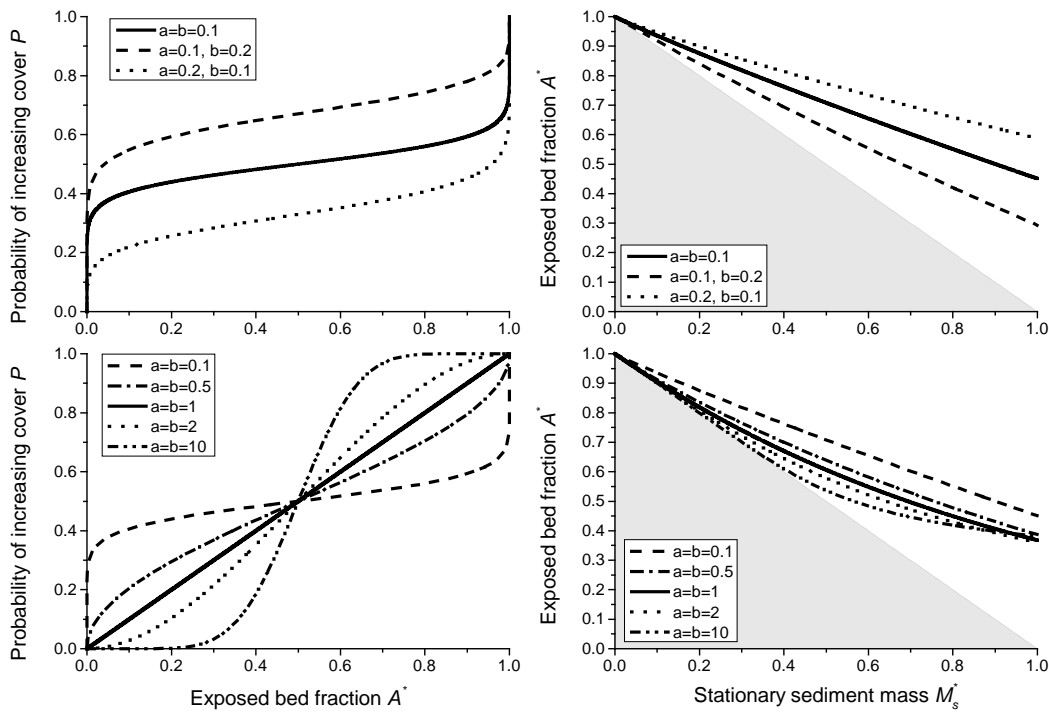


Fig. 3: Examples for the use of the regularized incomplete Beta function (eq. 11) to parameterize $P$,
using various values for the shape parameters $a$ and $b$. The choice $a = b = 1$ gives a dependence that
is equivalent to the exponential cover function. Grey shading depicts the area where the cover
function cannot run due to conservation of mass.

## 2.2 Example of application using model data

To illustrate how the framework can be used, we apply it to data obtained from the CA model developed by Hodge and Hoey (2012). The CA model reproduces the transport of individual sediment grains over a smooth bedrock surface. In each time step, the probability of a grain being entrained is a function of the number of neighboring grains. If five or more of the eight neighbouring cells contain grains then the grain has probability of entrainment $p_c$, otherwise it has probability $p_i$. In most model runs $p_c$ was set to a value less than that of $p_i$, thus accounting for the impact of sediment cover in decreasing local shear stress (through increased flow resistance) and increasing the critical entrainment shear stress for grains (via lower grain exposure and increased pivot angles). Thus, in the model, grain scale dynamics of entrainment are varied by adjusting the values of $p_i$ and $p_c$. This has a direct effect on the reach-scale distribution of cover, which is captured by our $P$-function (eq. 3).

The model is run with a domain that is 100 cells wide by 1000 cells long, with each cell having the same area as a grain. Up to four grains can potentially be entrained from each cell in a time step, limiting the maximum sediment flux. In each time step random numbers and the probabilities are used to select the grains that are entrained, which are then moved a step length of ten cells downstream and deposited. Model results are insensitive to the step length. A fixed number of grains are also supplied to the upstream end of the model domain. A smoothing algorithm is applied to prevent unrealistically tall piles of grains developing in cells if there are far fewer grains in adjacent cells. After around 500 time steps the model typically reaches a steady state condition in which the number of grains supplied to and leaving the model domain are equal. Sediment cover is measured in a downstream area of the model domain and is defined as grains that are not entrained in a given time step. Consequently grains that are deposited in one time step, and entrained in the following one do not contribute to the sediment cover, and so the model implicitly incorporates the effect of local sediment cover on grain deposition.

Model runs were completed with a six different combinations of $p_i$ and $p_c$: 0.95/0.95, 0.95/0.75, 0.75/0.10, 0.75/0.30, 0.30/0.30 and 0.95/0.05. These combinations were selected to cover the range of relationships between relative sediment supply $Q_s^*$ and the exposed bed fraction $A^*$ observed by Hodge and Hoey (2012). For each pair of $p_i$ and $p_c$ model runs were completed at least 20 different values of $Q_s^*$ in order to quantify the model behaviour.

Cover bed fraction and total mass on the bed produced by the model were converted using eq. (3) into the new probabilistic framework (Fig. 4). The derivative was approximated by simple linear finite differences, which, in the case of run-away alluviation, resulted in a non-continuous curve due to large gradients. The exponential benchmark (eq. 8) is also shown for comparison. The different model parameterisations produce results in which the probability of deposition on bedrock is both more and less likely than in the baseline case, with some runs showing both behaviours. Cases where the probability is more than the baseline case (i.e. grains are more likely to fall on uncovered areas) are associated with runs in which grains in clusters are relatively immobile. These runs are likely to be particularly affected by the smoothing algorithm that acts to move sediment from alluviated to bedrock areas. All model parameterisations predict greater bed exposure for a given normalised mass than is predicted by a linear cover relationship (Figure 3b). Runs with relatively more immobile cluster grains have a lower exposed fraction for the same normalised mass. Runs with low values of $p_i$ and $p_c$ seem to lead to behavior in which cover is more likely than in the exponential benchmark,

while for high values, it is less likely. However, these are complex interactions and it is difficult to
generalize the model behavior.

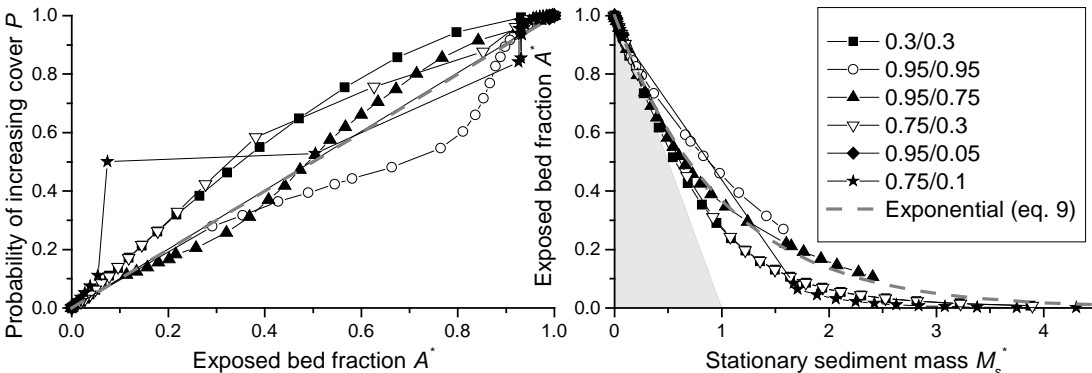

Fig. 4: Probability functions $P$ and cover function derived from data obtained from the model of
Hodge and Hoey (2012). The grey dashed line shows the exponential benchmark behavior. Grey
shading depicts the area where the cover function cannot run due to conservation of mass. The
legend gives values of the probabilities of entrainment $p_i$ and $p_c$ used for the runs (see text).
**3.   Cover development in time and space**
**3.1. Model derivation**
Previous descriptions of the cover effect relate the exposed fraction of the bed to the relative
sediment supply $Q_s^*$ (see eqs. 1 and 2). In this section, we derive a model to clarify the relationship
between the exposed fraction, $Q_s^*$ and $M_s$ and put it on a sound physical basis. To this end, the
probabilistic formulation introduced previously is extended to allow the calculation of the temporal
and spatial evolution of sediment cover in a stream. Here, we will derive the equations for the one
dimensional case (linear flume), but extensions to higher dimensions are possible in principle. The
derivation is inspired by the erosion-deposition framework (e.g. Charru et al., 2004), with some
necessary adaptions to make it suitable for channels with partial sediment cover (e.g., Turowski,
2009). In our system, we consider two separate mass reservoirs within a control volume. The first
reservoir contains all particles in motion, the total mass per bed area of which is denoted by $M_m$,
while the second reservoir contains all particles that are stationary on the bed, the total mass per
bed area of which is denoted by $M_s$. The reservoirs exchange mass by entrainment and deposition,
i.e., when a stationary particle is entrained it becomes mobile and when a mobile particle is
deposited, it becomes stationary. In addition to eq. 3, we need then three further equations, one to
connect the rate of change of mobile mass to the sediment flux in the flume, and one each to
describe mass conservation in the two reservoirs. Instead of the common approach tracking the
height of the sediment over a reference level, as is done in the classic mass conservation in fluvial
systems, the Exner equation (e.g. Paola and Voller, 2005), we use the total sediment mass on the bed
as a variable. Mobile sediment mass is supplied from upstream (Δin), leaves in the downstream
direction (Δout) and can be exchanged between the stationary and the mobile mass reservoirs by
entrainment ($E_{tot}$) and deposition ($D_{tot}$) (Fig. 5). The latter two parameters describe the exchange of
particles between reservoirs; in the single reservoir Exner equation these terms are not needed. It is
clear that for the problem at hand the choice of total mass or volume as a variable to track the
amount of sediment in the reach of interest is preferable to the height of the alluvial cover, since
necessarily, when cover is patchy, the height of the alluvium varies across the bed.

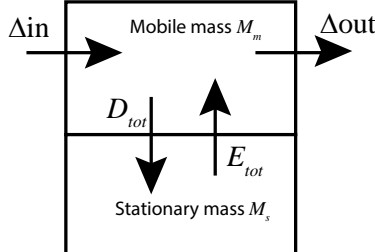

Fig. 5: Sediment dynamics at the bed are modelled by two reservoirs for stationary and mobile mass,
which can exchange material by entrainment ($E_{tot}$) and deposition ($D_{tot}$). Sediment mass can be
supplied from upstream ($\Delta$in) and can leave into the downstream direction ($\Delta$out).
The difference form of the mass balance for the mobile sediment is then given by (cf. Fig. 5)

$$\Delta M_m = (\Delta\text{in} - \Delta\text{out} + E_{tot} - D_{tot})\Delta t$$

(eq. 12)
Here, $\Delta M_m$ is the change in mobile sediment mass and $\Delta t$ is a change in time. As the length of a time
step is reduced to zero, a continuous version of eq. (12) is obtained, which reads

$$\frac{\partial M_m}{\partial t} = -\frac{\partial q_s}{\partial x} + E - D$$

(eq. 13)
Here, $x$ is the coordinate in the streamwise direction, $t$ the time, $q_s$ the sediment mass transport rate
per unit width, while $E$ is the mass entrainment rate per bed area and $D$ is the mass deposition rate
per bed area. Similarly, in the mass balance for the stationary mass reservoir, the rate of change of
the stationary sediment mass $M_s$ in time is the difference of the deposition rate $D$ and the
entrainment rate $E$:

$$\frac{\partial M_s}{\partial t} = D - E$$

(eq. 14)
It is useful to work with dimensionless variables by defining $t^* = t/T$ and $x^* = x/L$, where $T$ and $L$ are
suitable time and length scales, respectively. The dimensionless mobile mass per bed area $M_m^*$ is
equal to $M_m/M_0$, and eq. (13) becomes:

$$\frac{\partial M_m^*}{\partial t^*} = -\frac{\partial q_s^*}{\partial x^*} + E^* - D^*$$

(eq. 15)
Here,

$$q_s^* = \frac{T}{LM_0}q_s$$

(eq. 16)
The dimensionless entrainment and deposition rates, $E^*$ and $D^*$, are equal to $TE/M_0$ and $TD/M_0$,
respectively. Similarly, the balance for the stationary mass (eq. 14) can be written as

$$\frac{\partial M_s^*}{\partial t^*} = D^* - E^*$$

(eq. 17)
We also need sediment entrainment and deposition functions. The entrainment rate needs to be
modulated by the availability of sediment on the bed. If $M_s^*$ is equal to zero, no material can be
entrained. A plausible assumption is that the maximal entrainment rate, $E_{max}^*$, is equal to the
transport capacity.

$$E_{max}^* = q_t^*$$

(eq. 18)
Here, $q_t^*$ is the dimensionless mass transport capacity, which is related to the transport capacity per
unit width $q_t$ by a relation similar to eq. (16). To first order, the rate of change in entrainment rate,
$dE$, is proportional to the difference of $E_{max}$ and $E$, and to the rate of change in mass on the bed.

$$dE^* = (E_{max}^* - E^*)dM_s^* = (q_t^* - E^*)dM_s^*$$

(eq. 19)
Integrating, we obtain

$$E^* = E_{max}^*\left(1 - e^{-M_s^*}\right) = \left(1 - e^{-M_s^*}\right)q_t^*$$

(eq. 20)
Here, we used the condition $E^*(M_s^*=0) = 0$ to fix the integration constant to $E_{max}^*$. As required, eq.
(20) approaches $E_{max}^*$ as $M_s^*$ goes to infinity, and is equal to zero when $M_s^*$ is equal to zero. Using a
similar line of argument, and by assuming the maximum deposition rate to be equal to $q_s^*$, we arrive
at an equation for the deposition rate $D^*$.

$$D^* = \left(1 - e^{-M_m^*}\right)q_s^*$$

(eq. 21)
When $M_m^*$ is small, then the amount that can be deposited is limited by $M_m^*$. If $M_m^*$ is large, then
deposition is limited by sediment supply. Substituting eqs. (20) and (21) into eq. (17), we obtain:

$$\frac{\partial M_s^*(x^*,t^*)}{\partial t^*} = D^* - E^* = \left(1 - e^{-M_m^*(x^*,t^*)}\right)q_s^*(x^*,t^*) - \left(1 - e^{-M_s^*(x^*,t^*)}\right)q_t^*(x^*,t^*)$$

(eq. 22)
Note that $q_s^*/q_t^* = Q_s^*$. The equation for the mobile mass (eq. 14) becomes:

$$\frac{\partial M_m^*(x^*,t^*)}{\partial t^*} = -\frac{\partial q_s^*}{\partial x^*} - \left(1 - e^{-M_m^*(x^*,t^*)}\right)q_s^*(x^*,t^*) + \left(1 - e^{-M_s^*(x^*,t^*)}\right)q_t^*(x^*,t^*)$$

(eq. 23)
Finally, the sediment transport rate needs to be proportional to the mobile sediment mass times the
downstream sediment speed $U$, and we can write

$$q_s^*(x^*,t^*) = U^*(x^*,t^*)M_m^*(x^*,t^*)$$

(eq. 24)
Here
$$U^* = \frac{T}{L}U$$

(eq. 25)
After incorporating the original equation between $A^*$ and $M_s^*$ (eq. 3), the system of four differential
equations (3), (22), (23) and (24) contains four unknowns: the downstream gradient in the sediment
transport rate $\partial q_s^*/\partial x^*$, the exposed fraction of the bed $A^*$, the non-dimensional stationary mass $M_s^*$,
and the non-dimensional mobile mass $M_m^*$, while the non-dimensional transport capacity $q_t^*$ and the
non-dimensional downstream sediment speed $U^*$ are input variables, and $P$ is a externally specified
function. In addition, sediment input $q_s^*$ needs to be specified as an upstream boundary condition
and initial values for the mobile mass $M_m^*$ and the stationary mass $M_s^*$ need to be specified
everywhere.

**3.2. Time-independent solution**

In this chapter, we discuss the steady solution to the system of equations and thus clarify the
relationship between cover, stationary sediment mass, sediment supply and transport capacity.
Setting the time derivatives to zero, we obtain a time-independent solution, which links the exposed
area directly to the ratio of sediment transport rate to transport capacity. From eq. (23) it follows
that in this case, the entrainment rate is equal to the deposition rate and we obtain
$$\left(1 - e^{-\overline{M_m^*}}\right)\overline{q_s^*} = \left(1 - e^{-\overline{M_s^*}}\right)q_t^*$$

(eq. 26)
Here, the bar over the variables denotes their steady state value. Substituting eq. (24) to eliminate
$\overline{M_m^*}$ and solving for $\overline{M_s^*}$ gives

$$\overline{M_s^*} = -\ln\left\{1 - \left(1 - e^{-\overline{q_s^*}/U^*}\right)\frac{\overline{q_s^*}}{q_t^*}\right\} = -\ln\left\{1 - \left(1 - e^{-\frac{q_t^*\overline{Q_s^*}}{U^*Q_s^*}}\right)\overline{Q_s^*}\right\}$$

(eq. 27)
Note that we assume here that sediment cover is only dependent on the stationary sediment mass
on the bed and we thus neglect grain-grain interactions known as the dynamic cover (Turowski et al.,
2007). In analogy to eq. (24), we can write
$$q_t^* = U^* M_0^*$$

(eq. 28)
Here, $M_0^*$ is a characteristic dimensionless mass that depends on hydraulics and therefore implicitly
on transport capacity (which should not be confused with the minimum mass necessary to fully cover
the bed $M_0$). When sediment transport rate equals transport capacity, then $M_0^*$ is equal to the
mobile mass of sediment normalized by the reference mass $M_0$. It can be viewed as a proxy for the
transport capacity and is a convenient parameter to simplify the equations. The mobile mass can
then, in general, be written as follows (cf. Turowski et al., 2007), remembering that the relative
sediment supply $Q_s^* = 1$ when supply is equal to capacity:
$$M_m^* = M_0^* Q_s^*$$

(eq. 29)
If we use the exponential cover function (eq. 8) with eqs. (27), (28) and (29), we obtain

$$\overline{A^*} = 1 - \left(1 - e^{-\overline{q_s^*}/U^*}\right)\frac{\overline{q_s^*}}{q_t^*} = 1 - \left(1 - e^{-\frac{q_t^*\overline{Q_s^*}}{U^*Q_s^*}}\right)\overline{Q_s^*} = 1 - \left(1 - e^{-M_0^*\overline{Q_s^*}}\right)\overline{Q_s^*}$$

(eq. 30)
Similarly, equations can be found for the other analytical solutions of the cover function. For the
linear case (eq. 6), we obtain:
$$\overline{A^*} = 1 + \ln\left\{1 - \left(1 - e^{-M_0^*\overline{Q_s^*}}\right)\overline{Q_s^*}\right\}$$

(eq. 31)
For the power law case (eq. 9), we obtain:
$$\overline{A^*} = \left[1 + (1 - \alpha)\ln\left\{1 - \left(1 - e^{-M_0^*\overline{Q_s^*}}\right)\overline{Q_s^*}\right\}\right]^{\frac{1}{1-\alpha}}$$

(eq. 32)
The exponential cover function essentially leads to a combined linear and exponential relation
between $\overline{A^*}$ and $\overline{Q_s^*}$. Instead of a linear decline as the original linear cover model (eq. 1), or a
concave-up relationship as the original exponential model (eq. 2), the function is convex-up for all
solutions (Fig. 6). Adjusting $M_0^*$ shifts the lines: decreasing $M_0^*$ leads to a delayed onset of cover and
vice versa. The former result arises because a lower $M_0^*$ means that the sediment flux is conveyed
through a smaller mass moving at a higher velocity. The original linear cover function (eq. 1) can be
recovered from the exponential model with a high value of $M_0^*$, since the exponential term quickly

becomes negligible with increasing $\overline{Q_s^*}$ and the linear term dominates (Fig. 6C). Note that for the linear (eq. 5) and the power law cases (eq. 9), high values of $M_0^*$ may give $\overline{A^*} = 0$ for $\overline{Q_s^*} < 1$ (Fig. 6B,D), which is consistent with the concept of runaway alluviation. Using the beta distribution to describe $P$, a numerical solution is necessary, but a wide range of steady-state cover functions can be obtained (Fig. 7. By varying the value of $M_0^*$, an even wider range of behaviors can be obtained.

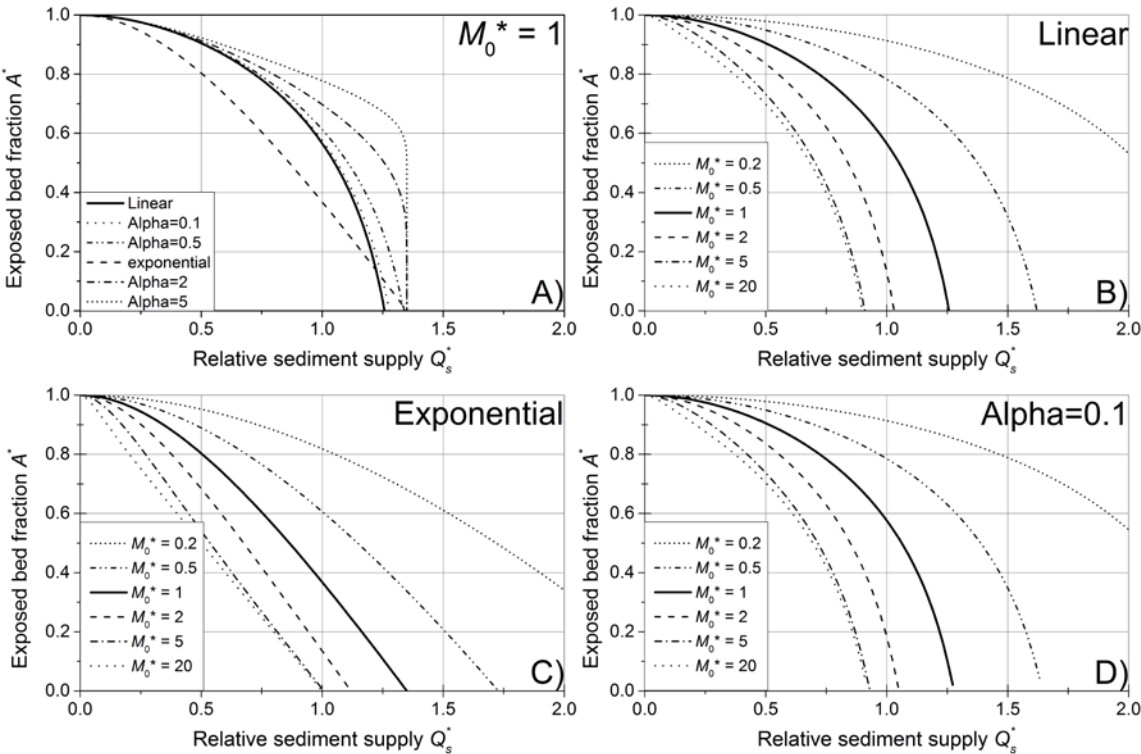

Fig. 6: Analytical solutions at steady state for the exposed fraction of the bed ($A^*$) as a function of relative sediment supply ($Q^*$, cf. Fig. 2). A) Comparison of the different solutions, keeping $M_0^*$ constant at 1. B) Varying $M_0^*$ for the linear case (eq. 31). C) Varying $M_0^*$ for the exponential case (eq. 30). D) Varying $M_0^*$ for the power law case with $\alpha = 0.1$ (eq. 32).

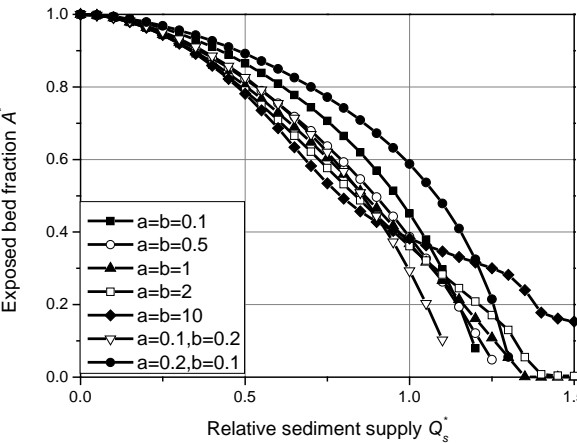

Fig. 7: Steady state solutions using the beta distribution to parameterize $P$ (eq. 10) for a range of parameters $a$ and $b$, and using $M_0^* = 1$ (cf. Fig. 3). The solutions were obtained by iterating the equations to a steady state, using initial conditions of $A^* = 1$ and $M_m^* = M_s^* = 0$.


The previous analysis shows that steady state cover is controlled by the characteristic dimensionless
mass $M_0{}^*$, which is equal to the ratio of dimensionless transport capacity and particle speed (eq. 28).
In the following, we relate $M_0{}^*$ to hydraulic variables and argue that it is, in general, not a constant.
Converting $M_0{}^*$ to dimensional variables, we can write
$$M_0^* = \frac{q_t^*}{U^*} = \frac{q_t}{M_0 U}$$

(eq. 33)
The minimum mass necessary to completely cover the bed per unit area, $M_0$, can be estimated
assuming a single layer of close-packed spherical grains residing on the bed (cf. Turowski, 2009),
giving:
$$M_0 = \frac{\pi \rho_s D_{50}}{3\sqrt{3}}$$

(eq. 34)
Here, $\rho_s$ is the sediment density and $D_{50}$ is the median grain size. We use equations derived by
Fernandez-Luque and van Beek (1976) from flume experiments that describe transport capacity and
particle speed as a function of bed shear stress (see also Lajeunesse et al., 2010, and Meyer-Peter
and Mueller, 1948, for similar equations):

$$q_t = 5.7 \frac{\rho_s \rho}{(\rho_s - \rho)g} \left( \frac{\tau}{\rho} - \frac{\tau_c}{\rho} \right)^{3/2}$$

(eq. 35)

$$U = 11.5 \left( \left( \frac{\tau}{\rho} \right)^{1/2} - 0.7 \left( \frac{\tau_c}{\rho} \right)^{1/2} \right)$$

(eq. 36)
Here, $\tau_c$ is the critical bed shear stress for the onset of bedload motion, $g$ is the acceleration due to
gravity and $\rho$ is the water density. Combining eqs. (34), (35) and (36) to get an equation for $M_0{}^*$ gives:

$$M_0^* = \frac{3\sqrt{3}}{2\pi} \frac{(\theta - \theta_c)^{3/2}}{\theta^{1/2} - 0.7\theta_c^{1/2}} = \frac{3\sqrt{3}\theta_c}{2\pi} \frac{(\theta/\theta_c - 1)^{3/2}}{(\theta/\theta_c)^{1/2} - 0.7}$$

(eq. 37)
Here, the Shields stress $\theta = \tau/(\rho_s - \rho)gD_{50}$, and $\theta_c$ is the corresponding critical Shields stress, and we
approximated 5.7/11.5 = 0.496 with 1/2 (compare to eqs. 35/36). At high $\theta$, when the threshold can
be neglected, eq. (37) reduces to a linear relationship between $M_0{}^*$ and $\theta$. Near the threshold, $M_0{}^*$ is
shifted to lower values as $\theta_c$ increases (Fig. 8). The systematic variation of $U^*$ with the hydraulic
driving conditions (eq. 36) implies that the cover function evolves differently in response to changes
in sediment supply and transport capacity. For a first impression, by comparing equations (35) and
(36), we assume that particle speed scales with transport capacity raised to the power of one third
(Fig. 9).

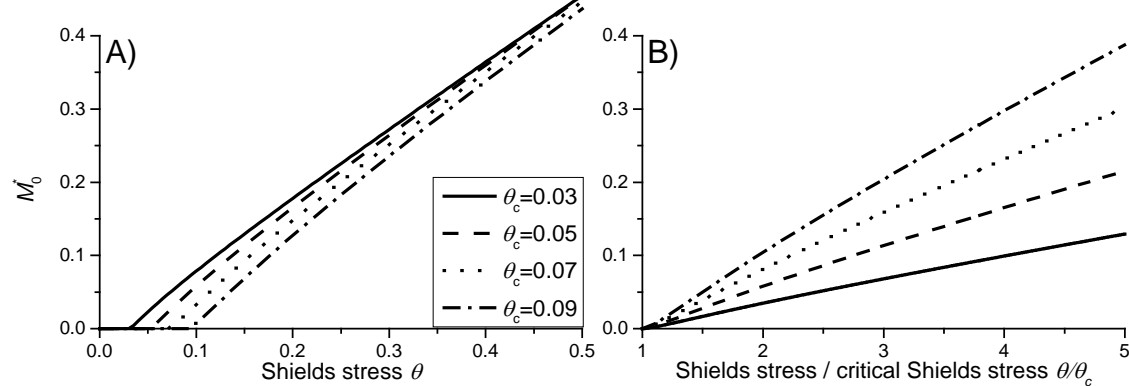

Fig. 8: The characteristic dimensionless mass $M_0{}^*$ depicted as a function of A) the Shields stress and B) the ratio of Shields stress to critical Shields stress (eq. 37).

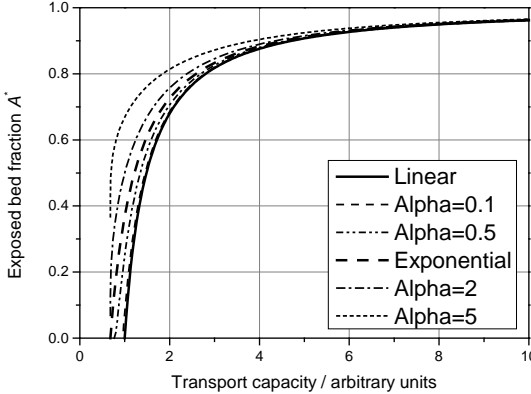

Fig. 9: Variation of the exposed bed fraction as a function of transport capacity, assuming that particle speed scales with transport capacity to the power of one third.

### 3.3 Temporal evolution of cover within a reach

To calculate the temporal evolution of cover on the bed within a single reach, we solved equations (3), (22), (23) and (24) numerically for a section of the bed with homogenous conditions using a simple linear finite difference scheme. In this case sediment input is a boundary condition, while sediment output, mobile and stationary sediment mass and the fraction of the exposed bed are output variables. In general, a change in sediment supply leads to a gradual adjustment of the output variables towards a new steady state (Fig. 10). It is desirable to obtain expressions for the response time of the system to external perturbation, such as a change in sediment supply or hydraulic conditions. Such a response time could then be compared to the time scales of changes in boundary conditions. For example, during a flood event, both transport capacity and sediment supply change over time. If these changes are slow in comparison of the response time of cover, the bed cover state can essentially keep up with the imposed changes at all times and therefore steady state equations (section 3.2) can be used to calculate its evolution. In contrast, if the imposed change is rapid in comparison to the response time, cover may lag behind and an approach that resolves cover as a dynamic variable is necessary. This may, for example, be important when studying the erosional behavior of channels in response to floods (see Lague, 2010; Turowski et al., 2013). Unfortunately, a general analytical solution is not possible, but results can be obtained for special cases. We first derive analytical solutions for the response time for a reach without upstream sediment supply and for a system responding to small perturbations in sediment supply or transport capacity (section

3.3.1) and discuss the system behavior (section 3.3.2). Finally, we apply the concepts to data from a
flood in a natural river and demonstrate that, for this specific case, because of the response times
the steady state relations do not capture cover behavior.
*3.3.1 System timescales*
First, consider a reach without upstream sediment supply, i.e., $q_s{}^* = 0$. Such a situation is rare in
nature, but could be easily created in flume experiments as a model test. Then, the time derivative of
stationary mass is given by:

$$\frac{\partial M_s^*}{\partial t^*} = -\left(1 - e^{-M_s^*}\right)q_t^*$$

(eq. 38)
Using the exponential cover model (eq. 8), we obtain:

$$\frac{1}{A^*(1 - A^*)}\frac{\partial A^*}{\partial t^*} = q_t^*$$

(eq. 39)
Equation (39) is separable and can be integrated to obtain

$$\ln(A^*) - \ln(1 - A^*) = t^* q_t^* + C$$

(eq. 40)
Letting $A^*(t^*=0) = A^*{}_0$, where $A^*{}_0$ is the initial cover, the final equation is in the form of a sigmoidal-
type function:

$$A^* = \frac{1}{1 + \left(\frac{1 - A_0^*}{A_0^*}\right)e^{-t^* q_t^*}}$$

(eq. 41)
By making the parameters in the exponent on the right hand side of eq. (42) dimensional, we get:

$$t^* q_t^* = \frac{t}{T}\frac{T}{LM_0}q_t = \frac{t q_t}{LM_0}$$

(eq. 42)
which allows a characteristic system time scale $T_E$ to be defined as

$$T_E = \frac{LM_0}{q_t}$$

(eq. 43)
Since this time scale is dependent on the transport capacity $q_t$, we can view it as a time scale
associated with the entrainment of sediment from the bed (cf. eq. 20) – hence the subscript $E$ on $T_E$.
From eq. (41), the exposed bed fraction evolves in an asymptotic fashion towards equilibrium
(Fig. 11). We can expect that there are other characteristic time scales for the system, for example
associated with sediment deposition or downstream sediment evacuation.
We can make some further progress and define a more general system time scale by performing a
perturbation analysis (Appendix A). For small perturbations in either $q_s{}^*$ or $q_t{}^*$, we obtain an
exponential term describing the transient evolution, which allows the definition of a system
timescale $T_S$

$$\exp\left\{-\left(\overline{q_t^*} - \left(1 - e^{-\overline{q_s^*}/\overline{U^*}}\right)\overline{q_s^*}\right)t^*\right\} = e^{-\frac{t}{T_S}}$$


(eq. 44)
Here, $\exp$ denotes the natural exponential function. The characteristic system time scale can then be
written as
$$T_S = \frac{LM_0}{\overline{q_t}\left(1 - \left(1 - e^{-\overline{q_s^*}/\overline{U^*}}\right)\frac{\overline{q_s}}{\overline{q_t}}\right)} = \frac{LM_0}{\overline{q_t}}e^{\overline{M_S^*}}$$

(eq. 45)
Note that for $q_s{}^* = 0$, eq. (45) reduces to eq. (43), as would be expected. Since $\overline{M_S^*}$ is directly related
to steady state bed exposure $\overline{A^*}$, we can rewrite the equation, for example by assuming the
exponential cover function (eq. 8), as
$$T_S = \frac{LM_0}{\overline{q_t}\,\overline{A^*}}$$

(eq. 46)
Since bed cover is more easily measurable than the mass on the bed, eq. (46) can help to estimate
system time scales in the field. Further, $\overline{A^*}$ varies between 0 and 1, which allows the estimation of a
minimum system time using eq. (43). As $\overline{A^*}$ approaches zero, the system time scale diverges.

To illustrate these additional dependencies, we have used numerical solutions of eqs. (3), (22), (23)
and (24) to calculate the time needed to reach 99.9% of total adjustment after a step change in
transport stage (chosen due to the asymptotic behavior of the system), analysed across a plausible
range of particle speeds $U$ (Fig. 12). Response time decreases as particle speed increases. This
reflects elevated downstream evacuation for higher particles speeds, resulting in a smaller mobile
particle mass and thus higher entrainment and lower deposition rates. Response time also increases
with increasing relative sediment supply $Q_s{}^*$. As the runs start with zero sediment cover, and the
extent of cover increases with $Q_s{}^*$, at higher $Q_s{}^*$ the adjusted cover takes longer to develop.

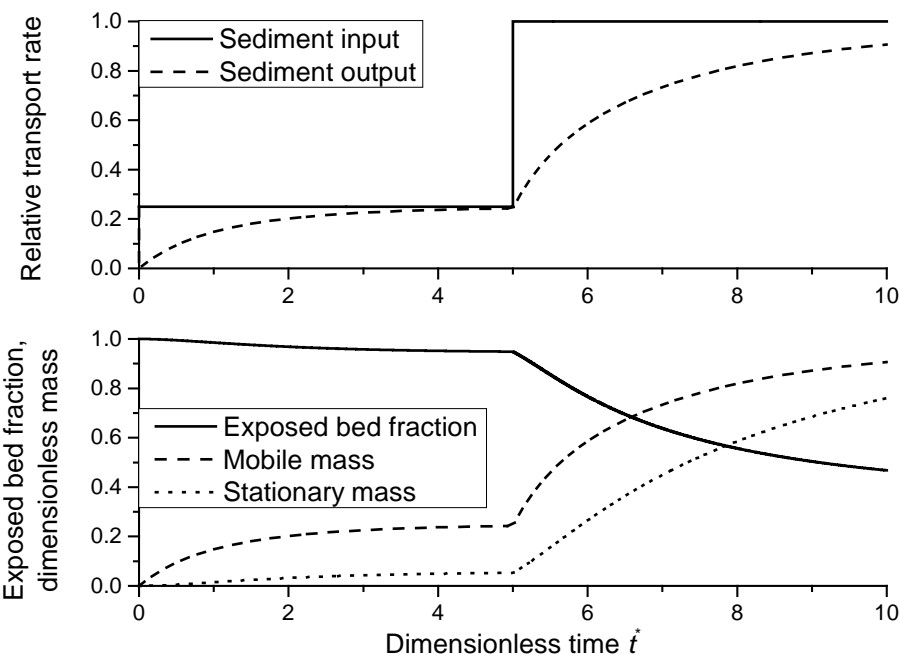

Fig. 10: Temporal evolution of cover for the simple case of a control box with sediment through-flux,
based on eqs. (3), (22), (23) and (24). Relative sediment supply (supply normalized by transport
capacity) was specified to 0.25 and increased to 1 at $t^* = 5$. The response of sediment output, mobile
and stationary sediment mass and the exposed bed fraction was calculated. Here, we used the
exponential function for $P$ (eq. 8) and $M_0^* = U^* = 1$. The initial values were $A^* = 1$ and $M_m^* = M_s^* = 0$.

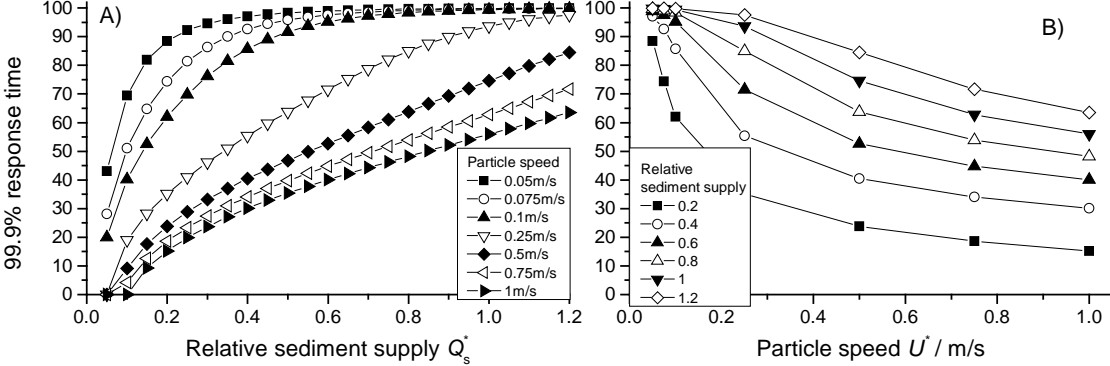

Fig. 11: Evolution of the exposed bed fraction (removal of sediment cover) over time starting with
different initial values of bed exposure, for the special case of no sediment supply, i.e., $q_s^* = 0$ (eq. 41)
and $q_t^* = 1$.

611

Fig. 12: Dimensionless time to reach 99.9% of the total adjustment in exposed area as a function of
A) transport stage and B) particle speed. All simulation were started with $A^* = 1$ and $M_m^* = M_s^* = 0$.

615
616

*3.3.2 Phase shift and gain in response to a cyclic perturbation*

The perturbation analysis (Appendix A) gives some insight into the response of cover to cyclic
sinusoidal perturbations. Let sediment supply be perturbed in a cyclic way described by an equation
of the form

$$q_s^* = \overline{q_s^*} + \delta q_s^* = \overline{q_s^*} + d \sin\left(\frac{2\pi t}{p}\right)$$

(eq. 47)

Here, the overbar denotes the temporal average, $\delta q_s^*$ is the time-dependent perturbation, $d$ is the
amplitude of the perturbation and $p$ its period. A similar perturbation can be applied to the transport
capacity (see Appendix A). The reaction of the stationary mass and therefore cover can then also be
described by sinusoidal function of the form (Appendix A)

$$\delta M_s^* = G \sin\left(\frac{2\pi t}{p} + \varphi\right)$$

(eq. 48)
Here, $\delta M_s^*$ is the perturbation of the stationary sediment mass around the temporal average, $G$ is
known as the gain, describing the amplitude response, and $\varphi$ is the phase shift. If the gain is large,
stationary mass reacts strongly to the perturbation; if it is small, the forcing does not leave a signal.
The phase shift is negative if the response lags behind the forcing and positive if it leads. The phase
shift can be written as
$$\varphi = \tan^{-1}\left(-2\pi\frac{T_S}{p}\right)$$

(eq. 49)
The gain can be written as
$$G = \frac{p}{T_S}\frac{Kd}{\sqrt{\left(\frac{p}{T_S}\right)^2 + 4\pi^2}}$$

(eq. 50)
Here, $d$ is the amplitude of the perturbation, and $K$ is a function of the time-averaged values of $q_s$, $q_t$
and $U$ and differs for perturbations in transport capacity and sediment supply (see Appendix A).
Thus, the system behavior can be interpreted as a function of the ratio of the period of perturbation
$p$ and the system time scale $T_s$. The period $p$ is large if the forcing parameter, i.e., discharge or
sediment supply, varies slowly and small when it varies quickly. According to eq. (49), the phase shift
is equal to $-\pi/2$ for low values of $p/T_s$ (quickly-varying forcing parameter), implying a substantial lag in
the adjustment of cover. The phase shift tends to zero as $p/T_s$ tends to infinity (Fig. 13). The gain
varies approximately linearly with $p/T_s$ for small $p/T_s$ (quickly-varying forcing parameter), while it is
approximately constant at a value of $Kd$ for large $p/T_s$ (slowly-varying forcing parameter) (eq. 50).
Thus, if the forcing parameter varies slowly, cover adjustment keeps up at all times.

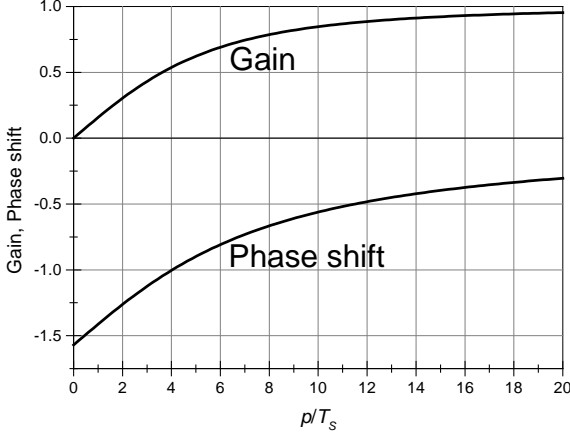

Fig. 13: Phase shift (eq. 49) and gain (eq. 50) as a function of the ratio of the period of perturbation $p$
and the system time scale $T_s$. For the calculation, the constant factor in the gain ($Kd$) was set equal to
one.

*3.3.3 A flood at the Erlenbach*
To illustrate the magnitude of the timescales using real data, we use a flood dataset from the
Erlenbach, a sediment transport observatory in the Swiss Prealps (e.g., Beer et al., 2015). There, near
a discharge gauge, bedload transport rates are measured at 1-minute resolution using the Swiss Plate
Geophone System, a highly developed and fully calibrated surrogate bedload measuring system (e.g.,
Rickenmann et al., 2012; Wyss et al. 2016). We use data from a flood on 20[th] June 2007 (Turowski et
al., 2009) with highest peak discharge that has so far been observed at the Erlenbach. The
meteorological conditions that triggered this flood and its geomorphic effects have been described in
detail elsewhere (Molnar et al., 2010; Turowski et al., 2009, 2013). The Erlenbach does not have a
bedrock bed in the sense that bedrock is exposed in the channel bed, however, the data provide a
realistic natural time series of discharge and bedload transport over the course of a single event.
Rather than predicting bed cover evolution for a natural system, for which we do not currently have
data for validation, we use the Erlenbach data to illustrate possible cover behavior during a fictitious
event with different initial sediment cover extents, using natural data to provide realistic boundary
conditions.
Using a median grain size of 80 mm, a sediment density of 2650 kg/m$^3$ and a reach length of 50 m,
we obtained $M_0$ = 128 kg/m$^2$. We calculated transport capacity using the equation of Fernandez
Luque and van Beek (1976). However, it is known that this and similar equations strongly
overestimate measured transport rates in streams such as the Erlenbach (e.g., Nitsche et al., 2011).
Consequently, we rescaled by setting the ratio of bedload supply to capacity to one at the highest
discharge. The exposed fraction was then calculated iteratively assuming $P = A^*$ (i.e., the exponential
cover formulation, eq. 8). In a real flood event, water discharge and sediment supply obviously do
not follow a small cyclic perturbation (Fig. 13). But we can tentatively relate the observations to the
theory by assuming that at each time step, the change in sediment supply can be represented by the
commencement of a sinusoidal perturbation with varying period. To estimate the effective period $p$,
one needs to take the derivatives of eq. (47).
$$\frac{dq_s^*}{dt} = \frac{d\delta q_s^*}{dt} = \frac{2\pi d}{p} \cos\left(\frac{2\pi t}{p}\right)$$
(eq. 51)
Setting $t$ = 0 for the time of interest, we can relate $p$ to the local gradient in bedload supply, which
can be measured from the data.

$$\frac{2\pi d}{p} = \frac{\Delta q_s^*}{\Delta t}$$
(eq. 52)
Assuming that all change in the response time is due to changes in the period (i.e., assuming a
constant amplitude, $d = 1$), we can obtain a conservative estimate of the range over which $p$ varies
over the course of an event.
$$p = 2\pi \frac{\Delta t}{\Delta q_s^*}$$
(eq. 53)
In the exemplary event, the evolution and final value of bed cover depends strongly on its initial
value (Fig. 14), indicating that the adjustment is incomplete. The system timescale is generally larger
than 1000 s and is inversely related to discharge via the dependence on transport capacity. The
$p/T_s$ ratio varies around one, with low values at the beginning of the flood and large values in the
waning hydrograph. Both the high values of the system time scale and the smooth evolution of bed
cover over the course of the flood imply that cover development cannot keep up with the variation in
the forcing characteristics. This dynamic adjustment of cover, which can lag forcing processes, may
thus play an important role in the dynamics of bedrock channels and probably needs to be taken into
account in modelling.

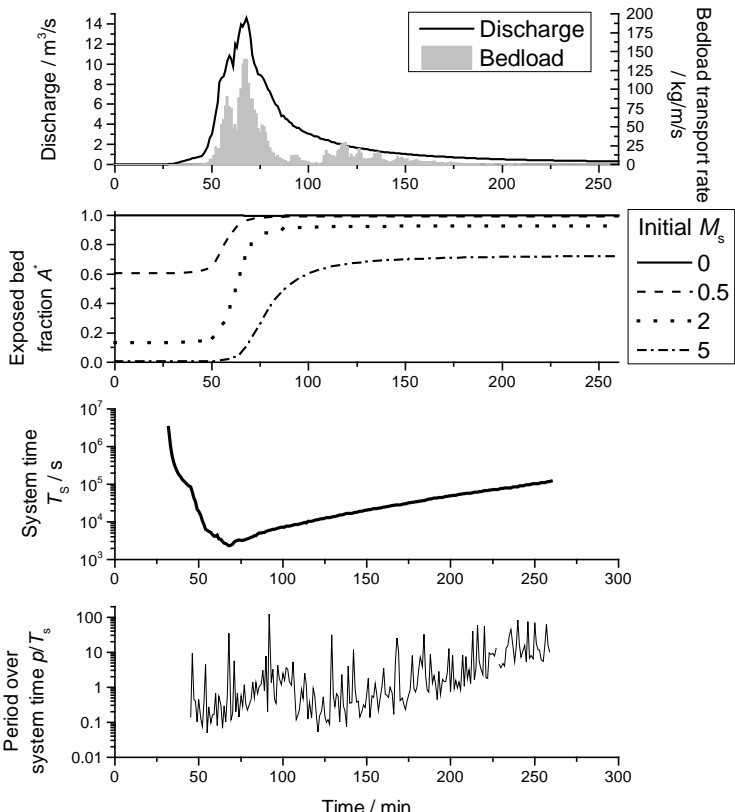

Fig. 14: Calculated evolution of cover during the largest event observed at the Erlenbach on 20th June
2007 (Turowski et al., 2009). Bedload transport rates were measured with the Swiss Plate geophone
sensors calibrated with direct bedload samples (Rickenmann et al., 2012). The final fraction of
exposed bedrock is strongly dependent on its initial value.

## 4.  Discussion
### 4.1 Model formulation
In principle, the framework for the cover effect presented here allows the formulation of a general
model for bedrock channel morphodynamics without the restrictions of previous models (e.g. Nelson
and Seminara, 2011; Zhang et al., 2015). To achieve this, the dependency of $P$ on various control
parameters needs to be specified. In general, $P$ should be controlled by local topography, grain size
and shape, hydraulic forcing, and the amount of sediment already residing on the bed. Furthermore,
the shape of the $P$ function should also be affected by feedbacks between these properties, such as
the development of sediment cover altering the local roughness and hence altering hydraulics and
local transport capacity (Inoue et al., 2014; Johnson, 2014). Within the treatment presented here, we
have explicitly accounted only for the impact of the amount of sediment already residing on the bed.
However, all of the mentioned effects can be included implicitly by an appropriate choice of $P$. The
exact relationships between, say, bed topography and $P$ need to be mapped out experimentally (e.g.,
Inoue et al., 2014), with theoretical approaches also providing some direction (cf. Johnson, 2014;
Zhang et al., 2015). Currently available experimental results (Chatanantavet and Parker, 2008;
Finnegan et al., 2007; Hodge and Hoey, 2016; Inoue et al., 2014; Johnson and Whipple, 2007) cover
only a small range of the possible parameter space and, in general, not all necessary parameters to
constrain $P$ were reported. Specifically the stationary mass of sediment residing on the bed is usually
not reported and can be difficult to determine experimentally, but is necessary to determine $P$.
Nevertheless, depending on the choice of $P$, our model can yield a wide range of cover functions that
encompasses reported functions both from numerical modelling (e.g., Aubert et al., 2016; Hodge and
Hoey, 2012; Johnson, 2014) and experiments (Chatanantavet and Parker, 2008; Inoue et al., 2014;
Sklar and Dietrich, 2001) (see Figs. 4 and 5).
The dynamic model put forward here is a minimum first order formulation, and there are some
obvious future alterations. We only take account of the static cover effect caused by immobile
sediment on the bed. The dynamic cover effect, which arises when moving grains interact at high
sediment concentration and thus reduce the number of impacts on the bed (Turowski et al., 2007),
could in principle be included into the formulation, but would necessitate a second probability
function specifically to describe this dynamic cover. It would also be possible to use different $P$-
functions for entrainment and deposition, thus introducing hysteresis into cover development. Such
hysteresis has been observed in experiments in which the equilibrium sediment cover was a function
of the initial extent of sediment cover (Chatanantavet and Parker, 2008; Hodge and Hoey, 2012).
Whether such alterations are necessary is best established with targeted laboratory experiments.
**4.2 Comparison to previous modelling frameworks**
We will briefly outline in this section the main differences to previous formulations of cover dynamics
in bedrock channels. Thus, the novel aspects of our formulation and the respective advantages and
disadvantages will become clear.
Aubert et al. (2015) coupled the movement of spherical particles to the simulation of a turbulent
fluid and investigated how cover depends on transport capacity and supply. Similar to what is
predicted by our analytical formulation, they found a range of cover function for various model set-
ups, including linear and convex-up relationships (compare the results in Fig. 6 to their Fig. 15).
Aubert et al. (2015) presented the so far most detailed physical simulations of bed cover formation
and the correspondence between the predictions is encouraging.
Nelson and Seminara (2011, 2012) formulated a morphodynamic model for bedrock channels. They
based their formulation on sediment concentration, which is in principle similar to our formulation
based on mass. However, Nelson and Seminara (2011, 2012) did not distinguish between mobile and
stationary sediment and linked local transport directly to sediment concentration. Further, Nelson
and Seminara (2011, 2012) assumed a direct correspondence between sediment concentration and
degree of cover, which is equivalent to the linear cover function (eq. 6). In this case, it is assumed
that grains are always deposited on uncovered bed and the different possible distributions of
particles within a grid node are not taken into account. Practically, this implies that the grid size
needs to be of the order of the grain size, because, strictly, the assumption is only valid if a single
grain can cover an entire grid node (cf. Fig. 1). Although different in various details, Inoue et al.
(2016) have used essentially the same approach as Nelson and Seminar (2011, 2012) to link bedload
concentration, transport and bed cover. Both of these models allow the 2D modelling of bedrock
channel morphology. Although we have not fully developed such a model in the present paper, our
model framework could easily be extended to 2D problems.
Inoue et al. (2014) formulated a 1D model for cover dynamics and bedrock erosion. There, they
distinguish between stationary and mobile sediment using an Exner equation to capture sediment
mass conservation. The degree of bed cover is related to transport rates and sediment mass via a
saturation volume, which is related to our characteristic mass $M_0^*$ (see section 3.2). A key difference
between Inoue et al.'s (2014) model and the one presented here lies in the sediment mass
conservation equations (eqs. 13 and 14), in which we explicitly take account of both entrainment and
deposition. In addition, with the function $P$, describing the relationship between deposited mass and
degree of cover, we provide a more flexible framework for complex simulations where the bed needs
to be discretized (e.g., 2D models or reach-scale formulations).
Zhang et al. (2015) formulated a bed cover model specifically for beds with macro-roughness. There,
deposited sediment always fills topographic lows from their deepest positions, such that there is a
reach-uniform sediment level. While the model provides a fundamentally different approach to what
is suggested here, its applicability is limited to very rough beds and the assumption of a sediment
elevation that is independent of the position on the bed seems physically unrealistic. In principle, the
probabilistic framework presented here should be able to deal with macro-rough beds, by making
the $P$-function (eq. 3) explicitly dependent on roughness, and thus allows a more general treatment
of the problem of bed cover.
Within this paper, we focused on the dynamics of bed cover, rather than on the modelling of the
dynamics of entire channels. The probabilistic formulation using the parameter $P$ provides a flexible
framework to connect the sediment mass residing on the bed with the exposed bedrock fraction.
This particular element has not been treated in any of the previous models and could be easily
implemented in other approaches dealing with sediment fluxes along and across the stream and the
interaction with erosion and, over long time scales, channel morphology. However, it is as yet
unclear how flow hydraulics, sediment properties and other conditions affect $P$ and this should be
investigated in targeted laboratory experiments.
**4.3 Further implications**
Based on field data interpretation, Phillips and Jerolmack (2016) argued that bedrock rivers adjust
such that, similar to alluvial channels, medium sized floods are most effective in transporting
sediment, and that channel geometry therefore can quickly adjust their transport capacity to the
applied load and therefore achieve grade (cf. Mackin, 1948). They conclude that bedrock channels
can adjust their morphologic parameters (channel width and shape) quickly in response to changing
boundary conditions. In contrast, our model suggests that instead bed cover can be adjusted to
achieve grade. In steady state, time derivatives need to be equal to zero. Thus, entrainment equals
deposition (eq. 14), implying that the downstream gradient in sediment transport rate is equal to
zero (eq. 13). When sediment supply or transport capacity change, the exposed bedrock fraction can
adjust to achieve a new steady state and a change of the channel geometry is unnecessary. These
changes in sediment cover can occur far more rapidly than changes in width and cross-sectional
shape (compare to eq. 46). Whether a steady state is achieved depends on the relative magnitude of
the timescales of perturbation and cover adjustment (see section 3). Our results imply that bedrock
channels have two distinct time scales to adjust to changing boundary conditions to achieve grade.
Over short times, bed cover is adjusted. This can occur rapidly. Over long time scales, channel width,
cross-sectional shape and slope are adjusted.
**5. Conclusions**
The probabilistic view put forward in this paper offers a framework into which diverse data on bed
cover, whether obtained from field studies, laboratory experiments or numerical modeling, can be
easily converted to be meaningfully compared. The conversion requires knowledge of the mass of
sediment on the bed and the evolution of exposed fraction of the bed. Within the framework,
individual data sets can be compared to the exponential benchmark and linear limit cases, enabling
physical interpretation. Furthermore, the formulation allows the general dynamic sub-grid modelling
of bed cover. Depending on the choice of $P$, the model yields a wide range of possible cover
functions. Which of these functions are appropriate for natural rivers and how they vary with factors
including topography needs to mapped out experimentally.
It needs to be noted here that the precise formulation of the entrainment and deposition functions
also affects steady state cover relations. When calibrating $P$ on data, it cannot always be decided
whether a specific deviation from the benchmark case results from varying entrainment and
deposition processes or from changes in the probability function driven for example by variations in
roughness. For the prediction of the steady state cover relations and for the comparison of data sets,
this should not matter, but the dynamic evolution of cover could be strongly affected.
The system timescale for cover adjustment is inversely related to transport capacity. This time scale
can be long and in many realistic situations, cover cannot instantaneously adjust to changes in the
forcing conditions. Thus, dynamic cover adjustment needs to be taken into account when modelling
the long-term evolution of bedrock channels.
Our model formulation implies that bedrock channels adjust bed cover to achieve grade. Therefore,
bedrock channel evolution is driven by two optimization principles. On short time scales, bed cover
adjusts to match the sediment output of a reach to its input. Over long time scales, width and slope
of the channel evolve to match long-term incision rate to tectonic uplift or base level lowering rates.

**Appendix A: Perturbation analysis**

Here, we derive the effect of a small sinusoidal perturbation of the driving variables, namely
sediment supply $q_s{}^*$ and transport capacity $q_t{}^*$, on cover development. The perturbation of the
driving variables can be written as

$$q_s^* = \overline{q_s^*} + \delta q_s^*$$


(eq. A1)

$$q_t^* = \overline{q_t^*} + \delta q_t^*$$

(eq. A2)
Here, the bar denotes the average of the quantity at steady state, while $\delta q_s{}^*$ and $\delta q_t{}^*$ denote the
small perturbation. The exposed area can be similarly written as

$$A^* = \overline{A^*} + \delta A^*$$


(eq. A3)
Steady state cover is directly related to the mass on the bed $M_s{}^*$ by eq. (3), which, as long as $P$ is
independent of time, we can rewrite as

$$\frac{dA^*}{dt} = -P\frac{dM_s^*}{dt}$$


(eq. A4)
Substituting eq. (A3) and a similar equation for $M_s{}^*$,

$$M_s^* = \overline{M_s^*} + \delta M_s^*$$


(eq. A5)
we obtain

$$\frac{d\delta A^*}{dt} = -P\frac{d\delta M_s^*}{dt}$$


(eq. A6)
Here, the averaged terms drop out as they are independent of time. If $P$ and the steady state
solution for $A^*$ are known, a direct relationship between $A^*$ and $M_s{}^*$ can be derived. For example, for
the exponential cover model (eq. 8), substituting eqs. (A3) and (A5), we find

$$\overline{A^*} + \delta A^* = e^{-\overline{M_s^*}-\delta M_s^*} = e^{-\overline{M_s^*}}e^{-\delta M_s^*} = \overline{A^*}e^{-\delta M_s^*} \approx \overline{A^*}(1 - \delta M_s^*)$$


(eq. A7)
Here, since the $\delta$ variables are small, we approximated the exponential term using a Taylor expansion
to first order. We obtain

$$\delta A^* = -\overline{A^*}\delta M_s^*$$


(eq. A8)
It is therefore sufficient to derive the perturbation solution for $M_s{}^*$, the time evolution of which is
given by eq. (22). Eliminating $M_m{}^*$ using eq. (24), we obtain

$$\frac{\partial M_s^*}{\partial t^*} = \left(1 - e^{-q_s^*/U^*}\right)q_s^* - \left(1 - e^{-M_s^*}\right)q_t^*$$


(eq. A9)

**Perturbation of sediment supply**

First, let us look at a perturbation of sediment supply $q_s{}^*$, while other parameters are held constant.
Substituting eq. (A1) and (A5) into (A9), we obtain

$$\frac{\partial \delta M_s^*}{\partial t^*} = \left(1 - e^{-\left(\overline{q_s^*}+\delta q_s^*\right)/U^*}\right)\left(\overline{q_s^*} + \delta q_s^*\right) - \left(1 - e^{-\overline{M_s^*}-\delta M_s^*}\right)q_t^*$$


(eq. A10)
Again, since the $\delta$ variables are small, we can replace the relevant exponentials with Taylor expansion
to first order:
$$e^{-\delta q_s^*/U^*} \approx 1 - \frac{\delta q_s^*}{U^*}$$

(eq. A11)
A similar approximation applies for the exponential in $M_s^*$. Substituting eq. (A11) into eq. (A10),
expanding the multiplicative terms, dropping terms of second order in the $\delta$ variables and
rearranging, we get
$$\frac{\partial \delta M_s^*}{\partial t^*} = \delta q_s^* \left(1 - e^{-\overline{q_s^*}/U^*} + \frac{\overline{q_s^*}}{U^*} e^{-\overline{q_s^*}/U^*}\right) - \delta M_s^* \left(q_t^* - \left(1 - e^{-\overline{q_s^*}/U^*}\right)\overline{q_s^*}\right)$$

(eq. A12)
The perturbation is assumed to be sinusoidal
$$\delta q_s^* = d \sin\left(\frac{2\pi t}{p}\right)$$

(eq. A13)
Here, $p$ is the period of the perturbation and $d$ is its amplitude. Note that, to be consistent with the
assumptions previously made, $d$ needs to be small in comparison with the average sediment supply.
Substituting, eq. (A12) can be integrated to obtain the solution
$$\delta M_s^* = G_{q_s^*} \sin\left(\frac{2\pi t}{P} + \varphi_{q_s^*}\right) + C\exp\left\{-\left(q_t^* - \left(1 - e^{-\overline{q_s^*}/U^*}\right)\overline{q_s^*}\right)\frac{t}{T}\right\}$$

where $C$ is a constant of integration. The gain is given by
$$G_{q_s^*} = \frac{p}{T} \frac{\left(1 - e^{-\overline{q_s^*}/U^*} + \frac{\overline{q_s^*}}{U^*} e^{-\overline{q_s^*}/U^*}\right)d}{\sqrt{\left(q_t^* - \left(1 - e^{-\overline{q_s^*}/U^*}\right)\overline{q_s^*}\right)^2 \left(\frac{p}{T}\right)^2 + 4\pi^2}}$$

(eq. A14)
And the phase shift by
$$\varphi_{q_s^*} = \tan^{-1}\left[-\frac{2\pi}{\frac{p}{T}\left(q_t^* - \left(1 - e^{-\overline{q_s^*}/U^*}\right)\overline{q_s^*}\right)}\right]$$

(eq. A15)

**Perturbation of transport capacity**

The perturbation of the transport capacity $q_t^*$ is a little more complicated, since both $q_t^*$ and $U^*$ are
explicitly dependent on hydraulics (e.g., shear stress; see eqs. 43 and 44), and thus $U^*$ is implicitly
dependent on $q_t^*$ and $\delta q_t^*$. To circumvent this problem, we expand the exponential term featuring
$U^*(\delta q_t^*)$ in eq. (A9) using a Taylor series expansion around $\delta q_t^* = 0$.

$$\exp\left\{-\frac{q_s^*}{U^*(\delta q_t^*)}\right\} \approx \exp\left\{-\frac{q_s^*}{U^*(\delta q_t^* = 0)}\right\}\left[1 - \frac{q_s^*}{U^{*2}(\delta q_t^* = 0)}\frac{\partial U^*}{\partial \delta q_t^*}(\delta q_t^* = 0)\delta q_t^*\right]$$

(eq. A16)
Both $U^*$ and its derivative are constants when evaluated at $\delta q_t^* = 0$. We can thus write

$$\exp\left\{-\frac{q_s^*}{U^*}\right\} = \exp\left\{-\frac{q_s^*}{\overline{U^*}}\right\}\left[1 - \frac{q_s^*}{\overline{U^*}^2}\overline{\left(\frac{\partial U^*}{\partial \delta q_t^*}\right)}\delta q_t^*\right] = [1 - C_0 \delta q_t^*]e^{-q_s^*/\overline{U^*}}$$


(eq. A17)
Here, $C_0$ is a constant. Proceeding as before by substituting eq. (A2), (A8) and (A17) into (A9),
expanding exponential terms containing $\delta$ variables, dropping terms of second order in the $\delta$
variables and rearranging, we obtain:

$$\frac{\partial \delta M_s^*}{\partial t^*} = \left( B q_s^* e^{-q_s^*/\overline{U^*}} + e^{-\overline{M_s^*}} - 1 \right) \delta q_t^* - \delta M_s^* \overline{q_t^*} e^{-\overline{M_s^*}}$$

(eq. A18)
A sinusoidal perturbation of the form

$$\delta q_t^* = d \sin\left(\frac{2\pi t}{p}\right)$$

(eq. A19)
yields the solution

$$\delta M_s^* = G_{q_t^*} \sin\left(\frac{2\pi t}{P} + \varphi_{q_t^*}\right) + C \exp\left\{-\left(\overline{q_t^*} - \left(1 - e^{-q_s^*/\overline{U^*}}\right)q_s^*\right)\frac{t}{p}\right\}\left\{-\left(\overline{q_t^*} - \left(1 - e^{-q_s^*/\overline{U^*}}\right)q_s^*\right)\frac{t}{T}\right\}$$

with

$$G_{q_t^*} = \frac{p}{T} \frac{\left(\frac{q_s^{*2}}{\overline{U^{*2}}}\overline{\left(\frac{\partial U^*}{\partial \delta q_t^*}\right)}e^{-q_s^*/\overline{U^*}} - \left(1 - e^{-q_s^*/\overline{U^*}}\right)\frac{q_s^*}{\overline{q_t^*}}\right)d}{\sqrt{\overline{q_t^*}^2 \left(\frac{p}{T}\right)^2 \left(1 - \left(1 - e^{-q_s^*/U^*}\right)\frac{q_s^*}{\overline{q_t^*}}\right)^2 + 4\pi^2}}$$

(eq. A20)
and

$$\varphi = \tan^{-1}\left(-\frac{2\pi}{\frac{p}{T}\left(\overline{q_t^*} - \left(1 - e^{-q_s^*/\overline{U^*}}\right)q_s^*\right)}\right)$$

(eq. A21)

**Summary**

Using the system timescale $T_S$, the phase shift and gain can be generally rewritten as

$$\varphi = \tan^{-1}\left(-2\pi \frac{T_S}{p}\right)$$

(eq. A22)

$$G = \frac{p}{T_S} \frac{Kd}{\sqrt{\left(\frac{p}{T_S}\right)^2 + 4\pi^2}}$$

(eq. A23)
Here, $K$ differs for perturbations in sediment supply and transport capacity, given by the equations

$$K_{q_s^*} = 1 - e^{-\overline{q_s^*}/U^*} + \frac{\overline{q_s^*}}{U^*}e^{-\overline{q_s^*}/U^*}$$

(eq. A24)

$$K_{q_t^*} = \frac{q_s^{*2}}{\overline{U^{*2}}}\overline{\left(\frac{\partial U^*}{\partial \delta q_t^*}\right)}e^{-q_s^*/\overline{U^*}} - \left(1 - e^{-q_s^*/\overline{U^*}}\right)\frac{q_s^*}{\overline{q_t^*}}$$

(eq. A25)


**Notation**

Overbars denote time-averaged quantities.

| | |
|---|---|
| $a$ | Shape parameter in the regularized incomplete Beta function. |
| $A^*$ | Fraction of exposed (uncovered) bed area. |
| $A_c^*$ | Fraction of covered bed area. |
| $b$ | Shape parameter in the regularized incomplete Beta function. |
| $B$ | Regularized incomplete Beta function. |
| $C$ | Constant of integration. |
| $C_0$ | Constant [m²s/kg]. |
| $d$ | Amplitude of perturbation [kg/m²s]. |
| $D$ | Sediment deposition rate per bed area [kg/m²s]. |
| $D_{tot}$ | Sediment deposition rate [kg/s]. |
| $D^*$ | Dimensionless sediment deposition rate. |
| $D_{50}$ | Median grain size [m]. |
| $e$ | Base of the natural logarithm. |
| $E$ | Sediment entrainment rate per bed area [kg/m²s]. |
| $E_{tot}$ | Sediment entrainment rate [kg/s]. |
| $E^*$ | Dimensionless sediment entrainment rate. |
| $E_{max}$ | Maximal possible dimensionless sediment entrainment rate. |
| $g$ | Acceleration due to gravity [m/s²]. |
| $G$ | Gain [kg/m²s]. |
| $I$ | Non-dimensional incision rate. |
| $k$ | Probability of sediment deposition on uncovered parts of the bed, linear implementation. |
| $k_I$ | Non-dimensional erodibility. |
| $K$ | Parameter in the gain equation. |
| $L$ | Characteristic length scale [m]. |
| $M_0$ | Minimum mass per area necessary to cover the bed [kg/m²]. |
| $M_0^*$ | Dimensionless characteristic sediment mass. |
| $M_m$ | Mobile sediment mass [kg/m²]. |
| $M_m^*$ | Dimensionless mobile sediment mass. |
| $M_s$ | Stationary sediment mass [kg/m²]. |
| $M_s^*$ | Dimensionless stationary sediment mass. |
| $p$ | Period of perturbation [s]. |
| $p_c$ | Probability of entrainment, CA model, blocked grains. |
| $p_i$ | Probability of entrainment, CA model, free grains. |
| $P$ | Probability of sediment deposition on uncovered parts of the bed. |
| $q_s$ | Mass sediment transport rate per unit width [kg/ms]. |
| $q_s^*$ | Dimensionless sediment transport rate. |
| $q_t$ | Mass sediment transport capacity per unit width [kg/ms]. |
| $q_t^*$ | Dimensionless transport capacity. |
| $Q_s^*$ | Relative sediment supply; sediment transport rate over transport capacity. |
| $Q_t$ | Mass sediment transport capacity [kg/s]. |
| $t$ | Time variable [s]. |
| $t^*$ | Dimensionless time. |
| $T$ | Characteristic time scale [s]. |

| 1007 | $T_E$ | Characteristic time scale for sediment entrainment [s]. |
|------|-------|---------------------------------------------------------|
| 1008 | $T_S$ | Characteristic system time scale [s]. |
| 1009 | $U$ | Sediment speed [m/s]. |
| 1010 | $U^*$ | Dimensionless sediment speed. |
| 1011 | $x$ | Dimensional streamwise spatial coordinate [m]. |
| 1012 | $x^*$ | Dimensionless streamwise spatial coordinate. |
| 1013 | $y$ | Dummy variable. |
| 1014 | $\alpha$ | Exponent. |
| 1015 | $\gamma$ | Fraction of pore space in the sediment. |
| 1016 | $\delta$ | denotes time-varying component. |
| 1017 | $\Delta$in | Sediment supply rate from upstream direction [kg/s]. |
| 1018 | $\Delta M_m$ | Change in mobile sediment mass [kg]. |
| 1019 | $\Delta$out | Transport rate of sediment leaving into the downstream direction [kg/s]. |
| 1020 | $\Delta t$ | Change in time [s]. |
| 1021 | $\theta$ | Shields stress. |
| 1022 | $\theta_c$ | Critical Shields stress. |
| 1023 | $\rho$ | Density of water [kg/m$^3$]. |
| 1024 | $\rho_s$ | Density of sediment [kg/m$^3$]. |
| 1025 | $\tau$ | Bed shear stress [N/m$^2$]. |
| 1026 | $\tau_c$ | Critical bed shear stress at the onset of bedload motion [N/m$^2$]. |
| 1027 | | |
| 1028 | | |

**Acknowledgements**

We thank J. Scheingross and J. Braun for insightful discussions and two anonymous reviewers and associate editor D. Egholm for their comments on the manuscript. The data from the Erlenbach is owned by and is used with permission of the Mountain Hydrology and Mass Movements Group at the Swiss Federal Research Institute for Forest Snow and Landscape Research WSL.

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

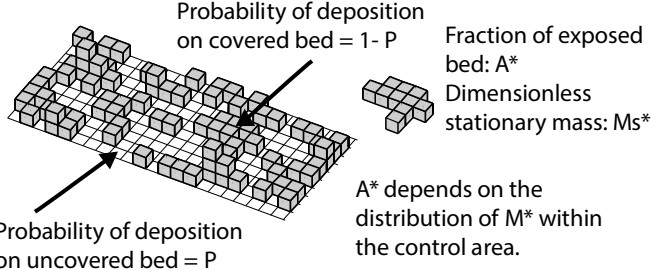

Fig. 1: Cartoon illustration of a bed partially covered by sediment. For purpose of illustration, the bed
is divided into a square raster, with each pixel of the size of a single grain. For a given number of
particles in the area of the bed of interest, the exposed area fraction of the bed is dependent on the
distribution of particles. Grains that sit on top of other grains do not contribute to cover. The
probability that a new grain is deposited on uncovered bed is given by $P$ (eq. 3).

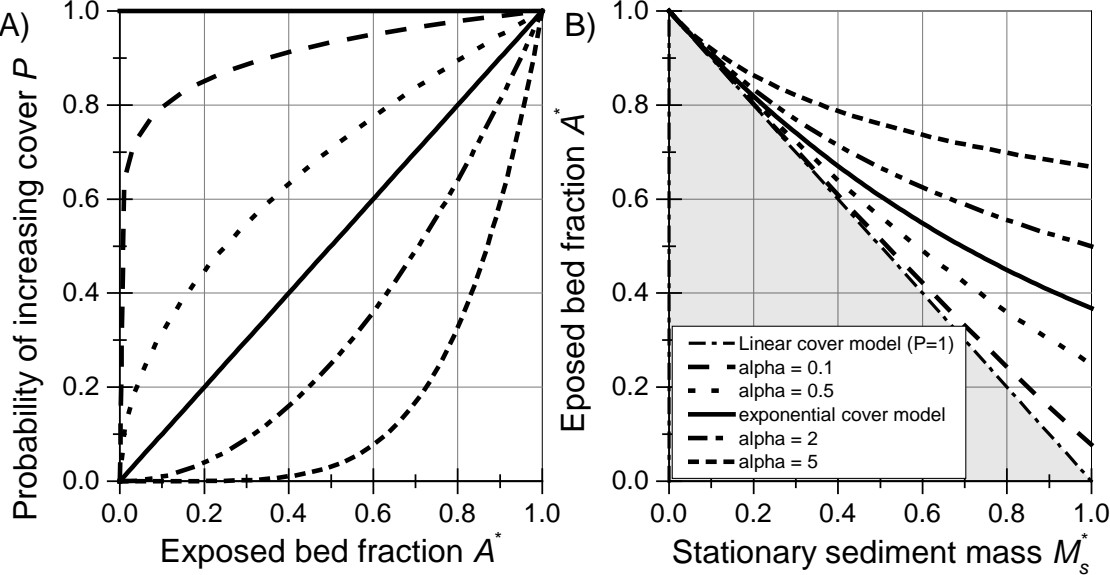

Fig. 2: A) Various examples for the probability function $P$ as a function of bedrock exposure $A^*$. B)
Corresponding analytical solutions for the cover function between $A^*$ and dimensionless sediment
mass $M_s^*$ using eq. (6), (7) and (9). Grey shading depicts the area where the cover function cannot
run due to conservation of mass.

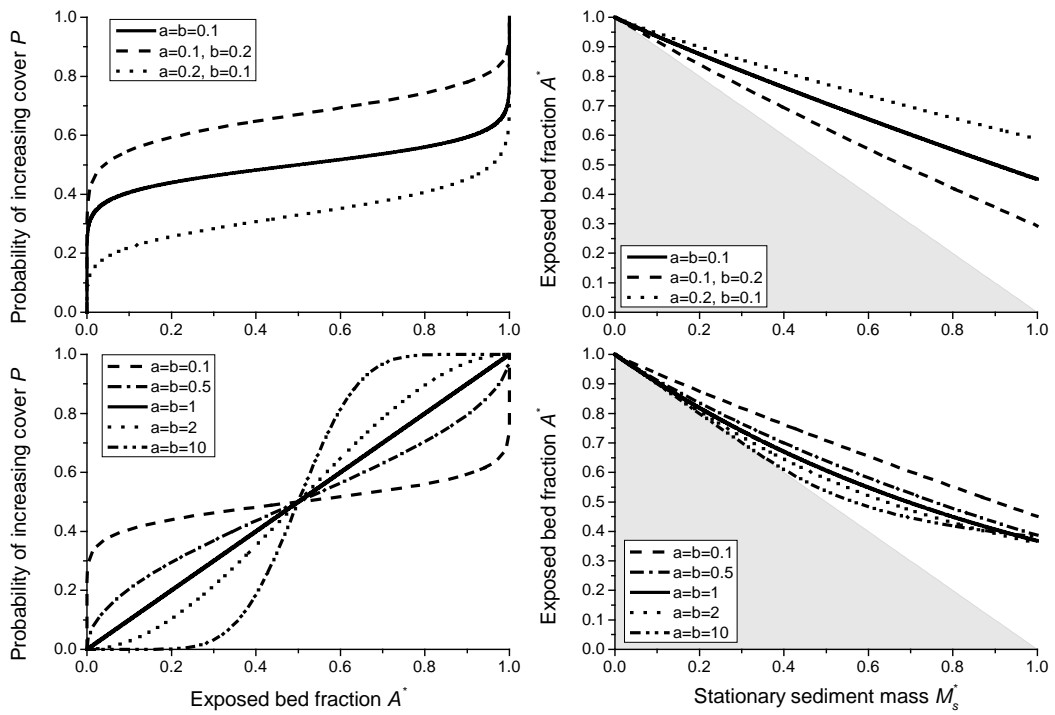

Fig. 3: Examples for the use of the regularized incomplete Beta function (eq. 11) to parameterize $P$,
using various values for the shape parameters $a$ and $b$. The choice $a = b = 1$ gives a dependence that
is equivalent to the exponential cover function. Grey shading depicts the area where the cover
function cannot run due to conservation of mass.

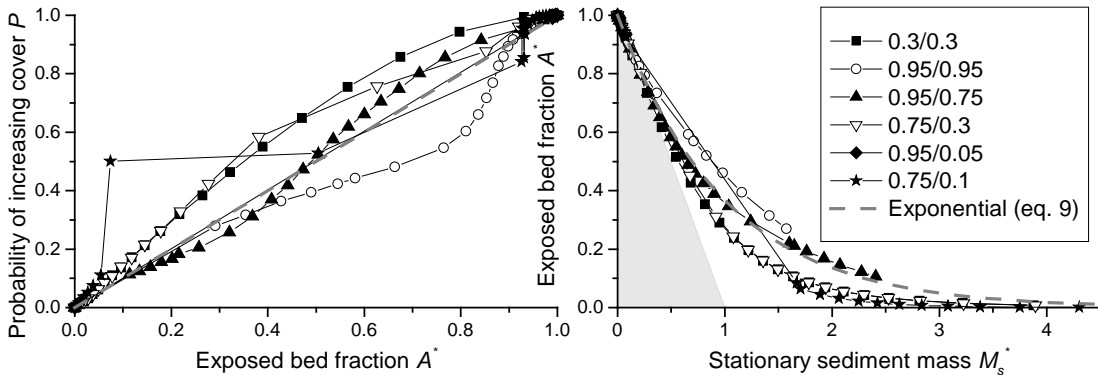

Fig. 4: Probability functions $P$ and cover function derived from data obtained from the model of
Hodge and Hoey (2012). The grey dashed line shows the exponential benchmark behavior. Grey
shading depicts the area where the cover function cannot run due to conservation of mass. The
legend gives values of $p_i$ and $p_c$ used for the runs (see text).

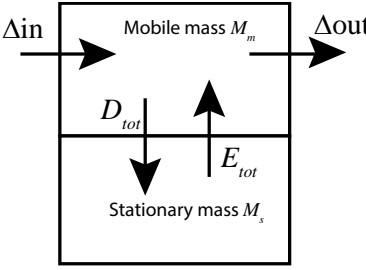

Fig. 5: Sediment dynamics at the bed are modelled by two reservoirs for stationary and mobile mass,
which can exchange material by entrainment ($E_{tot}$) and deposition ($D_{tot}$). Sediment mass can be
supplied from upstream ($\Delta$in) and can leave into the downstream direction ($\Delta$out).

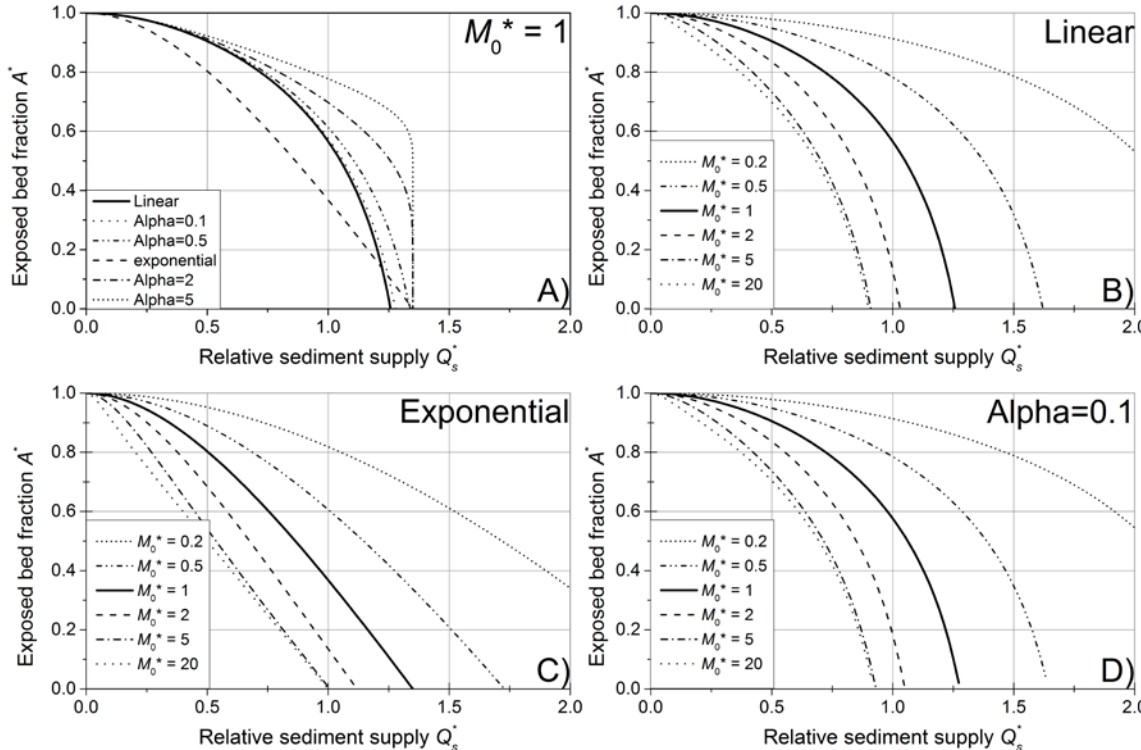

Fig. 6: Analytical solutions at steady state for the exposed fraction of the bed ($A^*$) as a function of
relative sediment supply ($Q^*$, cf. Fig. 2). A) Comparison of the different solutions, keeping $M_0^*$
constant at 1. B) Varying $M_0^*$ for the linear case (eq. 31). C) Varying $M_0^*$ for the exponential case (eq.
30). D) Varying $M_0^*$ for the power law case with $\alpha$ = 0.1 (eq. 32).

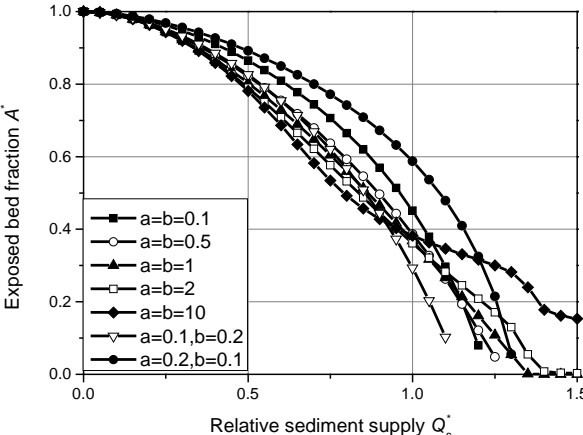

Fig. 7: Steady state solutions using the beta distribution to parameterize $P$ (eq. 10) for a range of
parameters $a$ and $b$, and using $M_0^* = 1$ (cf. Fig. 3). The solutions were obtained by iterating the
equations to a steady state, using initial conditions of $A^* = 1$ and $M_m^* = M_s^* = 0$.

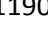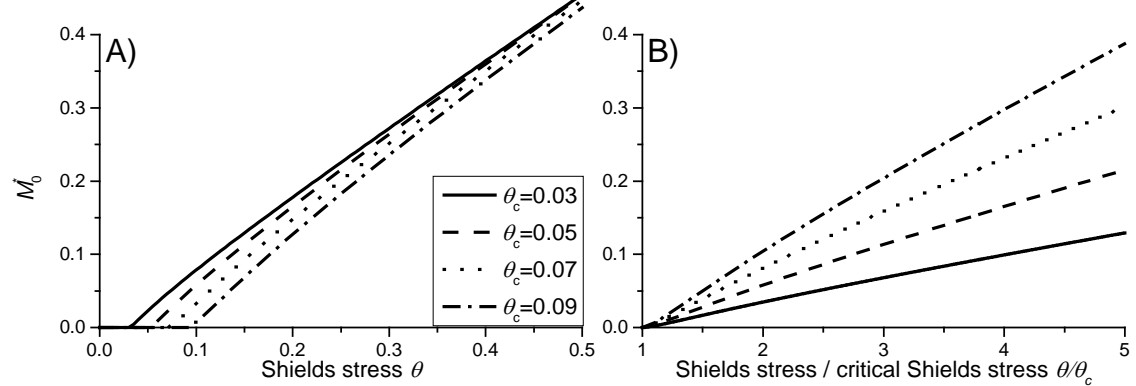

Fig. 8: The characteristic dimensionless mass $M_0^*$ depicted as a function of A) the Shields stress and
B) the ratio of Shields stress to critical Shields stress (eq. 37).

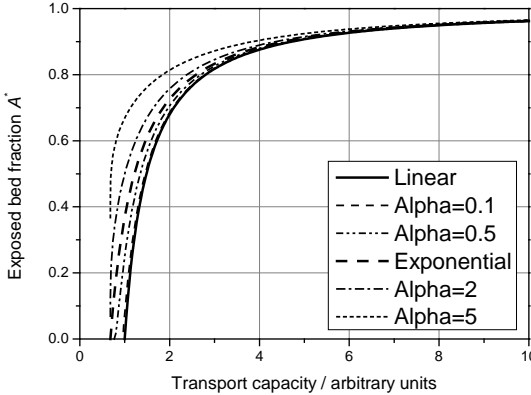

Fig. 9: Variation of the exposed bed fraction as a function of transport capacity, assuming that
particle speed scales with transport capacity to the power of one third.


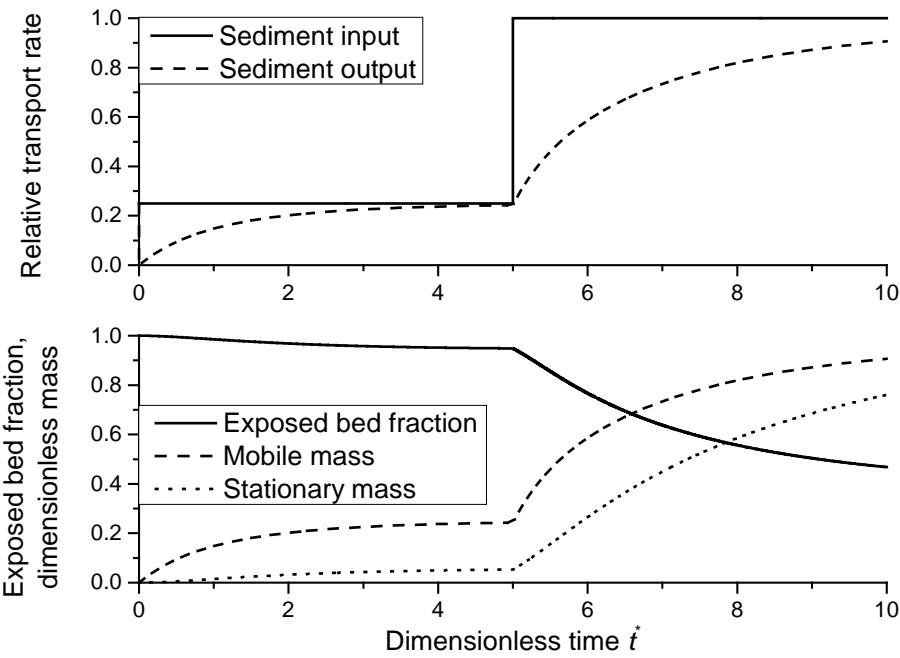

Fig. 10: Temporal evolution of cover for the simple case of a control box with sediment through-flux,
based on eqs. (3), (22), (23) and (24). Relative sediment supply (supply normalized by transport
capacity) was specified to 0.25 and increased to 1 at $t^* = 5$. The response of sediment output, mobile
and stationary sediment mass and the exposed bed fraction was calculated. Here, we used the
exponential function for $P$ (eq. 8) and $M_0^* = U^* = 1$. The initial values were $A^* = 1$ and $M_m^* = M_s^* = 0$.

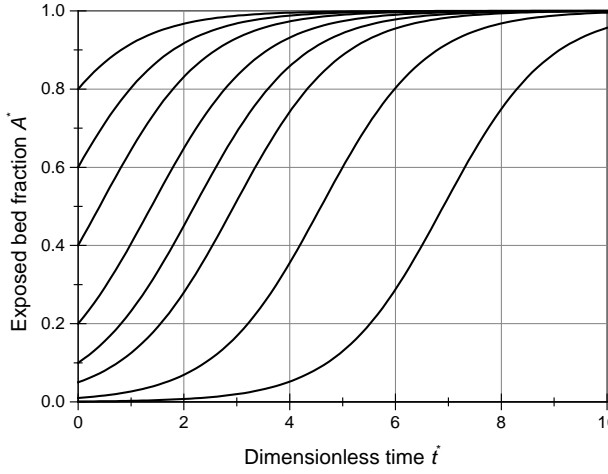

Fig. 11: Evolution of the exposed bed fraction (removal of sediment cover) over time starting with
different initial values of bed exposure, for the special case of no sediment supply, i.e., $q_s^* = 0$ (eq. 41)
and $q_t^* = 1$.

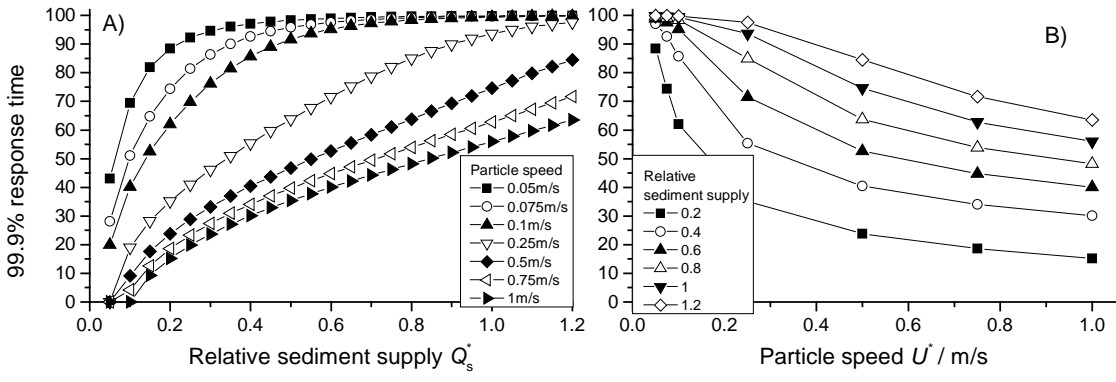

Fig. 12: Dimensionless time to reach 99.9% of the total adjustment in exposed area as a function of
A) transport stage and B) particle speed. All simulation were started with $A^* = 1$ and $M_m^* = M_s^* = 0$.

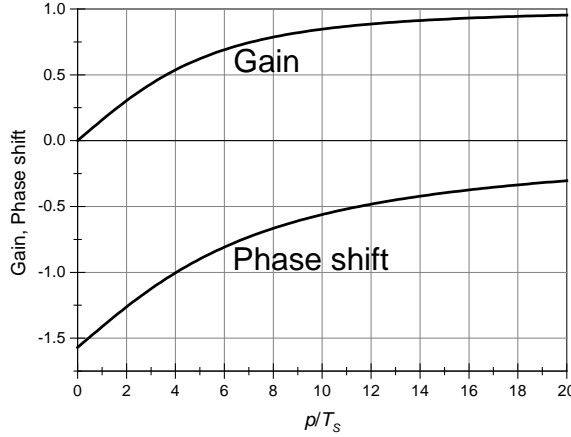

Fig. 13: Phase shift (eq. 49) and gain (eq. 50) as a function of the ratio of the period of perturbation $p$
and the system time scale $T_s$. For the calculation, the constant factor in the gain ($Kd$) was set equal to
one.

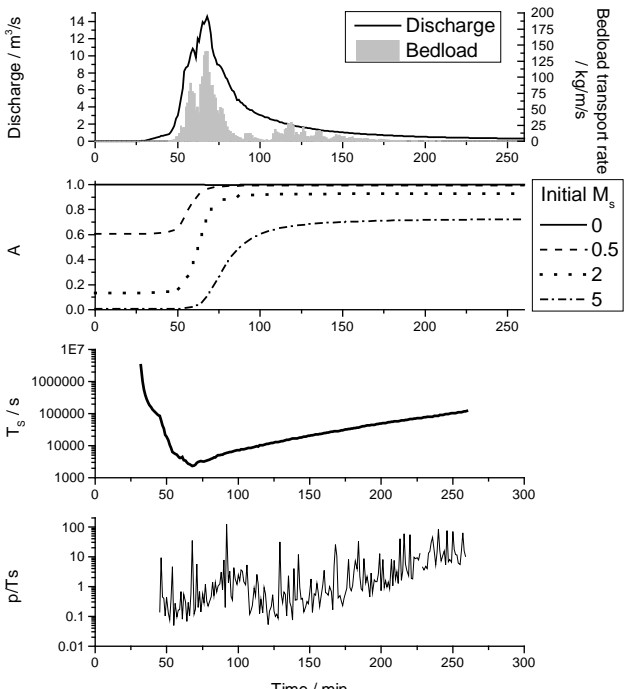

Fig. 14: Calculated evolution of cover during the largest event observed at the Erlenbach on 20[th] June
2007 (Turowski et al., 2009). Bedload transport rates were measured with the Swiss Plate geophone
sensors calibrated with direct bedload samples (Rickenmann et al., 2012). The final fraction of
exposed bedrock is strongly dependent on its initial value.