# Peer review of "A probabilistic framework for the cover effect in bedrock erosion"

_Earth Surface Dynamics, 2016_

## Referee Comment (RC1) · Anonymous Referee #1 · 17 Jan 2017

This paper emphasizes the need to think about how cover on a bedrock channel evolves, and I am supportive of that pursuit. I generally like the probabilistic approach of the paper. My review is not as deep as I would like it to be, because I got a bit lost in some of the details of the paper. I also had problems seeing how some of the sections tied together. Because the paper is so heavy in equations, and more importantly symbols, I think more reminders about what different symbols mean could make this a bit easier for the reader to follow. The table of symbols certainly helped. But anything the authors can do to improve the flow would be appreciated.

Line by line comments:

17 alleviated = alluviated?

Equation 3 : When I first read this, I thought "isn't this probability actually a function of

many things? Is there a reason that they only show it as A* and M_s*?" It's clear in the text that many variables are important, but I wondered why they were left out of the equation. Eventually I understood that the reason is because this paper focuses on A* and M_s*. Maybe this can be made clear from the start.

Section 2.1 in general – I know that the authors are not going to change this, but I had a very hard time remembering that A* is the fraction of exposed area, as in my head a cover function goes with fraction of area covered, not exposed. It's not that the authors aren't clear about the meaning, but somehow repeating the definition of A* more would have helped me. For example, I suggest that on all figure axes words accompany symbols, so the meaning of the variable is not mistaken (as I did many times.) I also had a hard time getting used to the meaning of P. In retrospect, after reading line 124 it is clear. However I wonder if this could be emphasized somehow. E.g. State what P is on fig 1A y-axis, or at least restate in the caption. I know repetition is frowned upon in scientific writing, but I need it in this paper.

L 159: I've never seen the word run used like this. Exist instead maybe?

Section 2.2: Again I'm not quite sure that you can do anything about this, but I got confused here because now you are talking about the probability of entrainment, in contrast to above which was the probability of deposition. Maybe just make sure this is clear to readers.

In this model all grains move the same length, right? So where they are deposited is not at all affected by whether or not there are grains in that location, right? This is confusing to me given that in the previous section deposition is probabilistic based on whether there are other grains present. So I had a hard time comparing this model with your framework.

Figure 3: Is this plotting the probability of deposition or entrainment? I think I know the answer, but maybe make this clear.

Equation 13 was hard for me. It seems like a new way to write this, but maybe you can walk me through it. Wouldn't E and D be included in d qs / dx? That is, wouldn't material be deposited if qs decreased downstream and entrained if qs increased downstream. I couldn't much evaluate this model because I didn't understand equation 13.

L 347: I think that this is $E^*(M^*_s = 0) = 0$, right? If so, maybe state explicitly.

L 417 – 419: I was confused about the comparison because you are not plotting the same thing as earlier plots. That is $Q^*$ is not $M\_s^*$, as was on the x-axes earlier. Can you help the reader relate these two variables? $Q^*$ is also not defined in your notation. Is $Q^*$ on the x-axes in figure 4 supposed to be $Q^*\_s$ with a bar over it? Maybe same for Figure 5 and also in the caption of figure 4?

L 465 Where did 5.7/11.5 come from?

Figure 10A – the x-axis is not formatted correctly.

Sentence starting on L 611: Does this mean that $A^* = 0$ at all times? I guess this gets to a point that I didn't understand in the previous section - when you are talking about the evolution of the mass on the bed through time, this could be under conditions of complete cover the entire time. It might be worth stating this directly. Although I'm not sure that I'm correct about this assumption. In Figure 12, assuming that A on the second from top plot is actually $A^*$, implies that there is not complete cover. I don't think I understood this section very well.

Also in this section, in a single flood event I'm having a hard time understanding what the meaning of p, or the period is. The previous section discussed the period in terms of a sine wave in sediment supply. But is that the case for this flood?

Sentence starting on L 669: This is great. Is this shown explicitly and I didn't catch it? If not, can you spell this out more directly?

L 678: This dynamics cover ... typo I think.

[Figure]

**ESurfD**

Interactive
comment

Sentence starting on Line 735 has a typo.

Section 4.3: The comparison with Philips and Jerolmack is a bit confusing to me. You state that in contrast to their findings, your findings suggest that bed cover is adjusted. But you didn't actually have channel morphology as a free variable. So I'm not sure how this study can contrast that one. I don't remember exactly, but I don't think they talked about sediment cover. So is this a fair way to compare the two studies?

I like how your conclusions stated this issue - both cover and channel morphology evolve. This makes sense to me, but the discussion in section 4.3 did not.

---

## Author Comment (AC1) · 17 Jan 2017

We thank the reviewer for the valuable comments, and in revisions, we will strive to clarify the rationale of the paper and the writing as best as possible. We will also be even more careful to avoid typos in the equations and in the symbols used.

Here, we want to briefly comment on eq. 13 that the reviewer found hard to understand. This is a mass balance that is probably more easily understood when viewed in a discrete framework. Consider a mass balance for a control volume in the river (Fig. 1). The rate of change of mobile mass per time, $\Delta M_m/\Delta t$ is then the sum of four terms: the mass influx per time from upstream $\Delta$in, the mass outflux per time downstream $\Delta$out, the entrained mass per time E and the deposited mass per time D. Both outflux $\Delta$out and deposition D reduce mass in the control volume and are therefore negative.

Thus:

$\Delta Mm/\Delta t = \Delta in - \Delta out + E - D$

(eq. 1)

Our equation 13 is essentially a version of this equation where the long-stream variable and the time are treated as continuous corresponding to the limit of infinitesimal length of the control volume and infinitesimal time steps. In this limit the term $\Delta in - \Delta out$ becomes dq_s/dx and the term $\Delta Mm/\Delta t$ becomes dM/dt.

Note that this is essentially equivalent to the standard Exner equation, which is written in terms of bed height rather than mass, and thus wraps both our reservoirs (mobile and stationary) into a single equation. Since we are explicitly interested in the mass on the bed, and since bed height necessarily varies across a partially covered bed, we opted for using mass as a main variable.

Fig. 1: Cartoon illustration of a control volume in the river. The mass balance for the mobile mass Mm leads to eq. 1.

**Fig. 1.** Cartoon illustration of a control volume in the river. The mass balance for the mobile mass Mm leads to eq. 1.

---

## Referee Comment (RC2) · Anonymous Referee #2 · 7 Mar 2017

In this manuscript, the authors presented a probabilistic framework for predicting partial cover in mixed bedrock-alluvial channels, which they used to explore how probability of sediment deposition, relative sediment supply, and particle speed interact. It represents a next step in the progress that has been made over the past several years in modeling areal fraction of sediment cover. Overall, I found this to be a good paper, with sound methods and interesting results.

1. My main concern with the manuscript is the equations for entrainment rate and deposition rate. Although the authors explain that eq. 20 approaches $E_{max}^*$ as $M_s^*$ goes to infinity, does the same apply to eq. 21? I think $M_m^*$ cannot be infinity because it is limited by the capacity value $M_0^*$. If so, when $M_s^*$ is very large, it is impossible to balance D with E.

2. The authors assume that increasing sediment deposition decreases local shear stress and increases the critical entrainment shear stress for grains (Line 186-192 and 242-245). I think this assumption is limited to the case of smooth bedrock. Sediment deposition does not necessarily decrease the flow velocity. In rough bedrock, increasing sediment deposition increases local shear stress and decreases the critical shear stress for grains.

3. In section 4.2, although the authors explain the differences from the model presented by Nelson and Seminara (2011, 2012); the model presented in this paper has more similarities to the model presented by Inoue et al. (2014). In the mentioned paper, they have distinguished between mobile and stationary sediment, and have not assumed a direct correspondence between sediment concentration and degree of cover, which is different from Nelson and Seminara (2011, 2012). I think the main difference is in the sediment continuity equation including entrainment rate and deposition rate. This equation seems very useful. I encourage the authors to explain the differences from Inoue et al. (2014) and the advantages of this sediment continuity equation.

Additional comments by line number below:

Line 140: There are situations when sediment does not accumulate even if the exposed part is zero. For example, runaway alluviation in Chatanantavet and Paker (2008).

Line 242-245: It is good to describe a physical reason for smoothing.

Line 345: Can $(1-e^{-Ms})q_t^*$ be converted to $(1-A)q_t^*$? If so, the entrainment rate is proportional to the areal fraction of sediment cover.

Figure 4 and Figure 5: $Q^* = Q_s^*$?

Figure 7: What is arbitrary unit? Is the sediment supply rate specified?

Figure 8: Please explain the distance from upstream end to downstream end, transport capacity, bed slope and grain size.

Figure10: Which equations are used to calculate 99% response time?

[Figure]

---

## Author Response (AR1)

We thank the reviewers for their valuable comments. Below we restate the comments in normal font and give our reply in *italics*. We hope that we have clarified everything and satisfactorily answered all the queries.

Reviewer #1
This paper emphasizes the need to think about how cover on a bedrock channel evolves, and I am supportive of that pursuit. I generally like the probabilistic approach of the paper. My review is not as deep as I would like it to be, because I got a bit lost in some of the details of the paper. I also had problems seeing how some of the sections tied together. Because the paper is so heavy in equations, and more importantly symbols, I think more reminders about what different symbols mean could make this a bit easier for the reader to follow. The table of symbols certainly helped. But anything the authors can do to improve the flow would be appreciated.
*We agree that we have many equations, but we do not think their number can be reduced without losing necessary mathematical detail (which would make the paper even more difficult to read to those who are only mildly enthusiastic about math…). In revisions, we have tried to clarify were possible.*

Line by line comments:
alleviated = alluviated?
*Corrected.*

Equation 3 : When I first read this, I thought "isn't this probability actually a function of many things? Is there a reason that they only show it as A* and M_s*?" It's clear in the text that many variables are important, but I wondered why they were left out of the equation. Eventually I understood that the reason is because this paper focuses on A* and M_s*. Maybe this can be made clear from the start.
*We are not sure how to take this comment. We meant to indicate the possible dependence on other variables with the three little dots within the equation. The next sentence in the text makes explicit what we meant by this and gives a list of possible control parameters. We are not sure how to improve clarity here. Maybe the reviewer or the editor have a specific idea? [Text edited to try and clarify further.]*

Section 2.1 in general – I know that the authors are not going to change this, but I had a very hard time remembering that A* is the fraction of exposed area, as in my head a cover function goes with fraction of area covered, not exposed. It's not that the authors aren't clear about the meaning, but somehow repeating the definition of A* more would have helped me. For example, I suggest that on all figure axes words accompany symbols, so the meaning of the variable is not mistaken (as I did many times.) I also had a hard time getting used to the meaning of P. In retrospect, after reading line 124 it is clear. However I wonder if this could be emphasized somehow. E.g. State what P is on fig 1A y-axis, or at least restate in the caption. I know repetition is frowned upon in scientific writing, but I need it in this paper.
*We try to improve readability in the revisions by repeating the definitions of variables. In particular, we will revise the figures to give both a verbal description of the parameters and their symbol.*
*Nevertheless, we want to stress here that the exposed fraction, as we have used it, and not the covered fraction is the parameter that is commonly used in equations, because erosion rate is proportional to the exposed fraction.*

L 159: I've never seen the word run used like this. Exist instead maybe?
*Changed as suggested.*

Section 2.2: Again I'm not quite sure that you can do anything about this, but I got confused here because now you are talking about the probability of entrainment, in contrast to above which was the probability of deposition. Maybe just make sure this is clear to readers.
*We have added qualifiers to make the relations more clear. To further avoid confusion, we have also changed to small font and added the symbols pi and pc to the notation list. We also added a sentence to the introductory paragraph, explaining the relation between pc and pi and the P-function.*

In this model all grains move the same length, right? So where they are deposited is not at all affected by whether or not there are grains in that location, right? This is confusing to me given that in the previous section deposition is probabilistic based on whether there are other grains present. So I had a hard time comparing this model with your framework.
*In the model sediment cover is only calculated for grains that are stationary for a time step; grains that are deposited and then immediately re-entrained do not count. Consequently the model implicitly incorporates the effect of local sediment cover on grain deposition. Note also that our P-function quantifies deposition on a reach-scale, while the formulation in the CA model is concerned with the grain scale. When grain scale dynamics are varied, this has a direct effect on the reach-scale, which is expressed in a different P-function. We have added some clarifying sentences.*

Figure 3: Is this plotting the probability of deposition or entrainment? I think I know the answer, but maybe make this clear.
*This is, of course, the probability P defined in eq. (3). We have tried to clarify.*

Equation 13 was hard for me. It seems like a new way to write this, but maybe you can walk me through it. Wouldn't E and D be included in d qs / dx? That is, wouldn't material be deposited if qs decreased downstream and entrained if qs increased downstream. I couldn't much evaluate this model because I didn't understand equation 13.
*We have explained the meaning of this equation in our brief reply in the discussion of the paper, which is reproduced below. We are at a little loss as how to deal with this comment. On the one hand, we can see that the reviewer was confused by this particular equation and as a result had a hard time understanding the remainder of the paper. On the other, the equation is a standard mass balance (derived from mass conservation), which is routinely used in river dynamics (in a slightly different form as Exner equation), geochemistry and many other disciplines. We think we have made these connections sufficiently clear in the introductory sentences preceding the presentation of the equation. Since a presentation of the derivation, including a cartoon, would take about half a page, we decided to not change the manuscript here. Nevertheless, we are willing to do so if the editor sees the necessity. Some of the confusion may arise because here we have two different reservoirs (Mm and Ms) and so we need to explicitly represent the transitions between them, whereas in a classic Exner the transition of sediment between bedload and bed is assumed and not explicitly represented. That is, if there is a downstream decrease in bedload, in the Exner equation the excess sediment is assumed to be deposited, while we make the exchange between mobile and stationary particle reservoirs explicit. Apart from the obvious advantage when one is interested in the size of the stationary reservoir, which is related to bed cover, the role of particle speed becomes also explicit. We have added a sentence to clarify this relation..*

*From our previous discussion: Here, we want to briefly comment on eq. 13 that the reviewer found hard to understand. This is a mass balance that is probably more easily understood when viewed in a discrete framework. Consider a mass balance for a control volume in the river (Fig. 1). The rate of change of mobile mass per time, ΔMm/Δt is then the sum of four terms: the mass influx per time from upstream Δin, the mass outflux per time downstream Δout, the entrained mass per time E and the deposited mass per time D. Both outflux Δout and deposition D reduce mass in the control volume and are therefore negative. Thus:*

*ΔMm/Δt = Δin − Δout +E − D*
*(eq. 1)*
*Our equation 13 is essentially a version of this equation where the long-stream variable and the time are treated as continuous corresponding to the limit of infinitesimal length of the control volume and infinitesimal time steps. In this limit the term Δin − Δout becomes dq_s/dx and the term ΔMm/Δt becomes dM/dt.*
*Note that this is essentially equivalent to the standard Exner equation, which is written in terms of bed height rather than mass, and thus wraps both our reservoirs (mobile and stationary) into a single equation. Since we are explicitly interested in the mass on the bed, and since bed height necessarily varies across a partially covered bed, we opted for using mass as a main variable.*

*Fig. 1: Cartoon illustration of a control volume in the river. The mass balance for the mobile mass Mm leads to eq. 1.*

[Figure]

L 347: I think that this is E*(M*s = 0) = 0, right? If so, maybe state explicitly.
*Changed as requested.*

L 417 – 419: I was confused about the comparison because you are not plotting the same thing as earlier plots. That is Q* is not M_s*, as was on the x-axes earlier. Can you help the reader relate these two variables?
This is correct. A main point of this chapter is to establish the relation between mass resevoirs Mm and Ms and the relative sediment supply Qs*. This partly has historical reasons – the cover function has so far been expressed in terms of Qs* (see, for example, Sklar and Dietrich, 2004), while experiments give the relation to Ms (see, for example, Sklar and Dietrich, 2001). The transformation between these two variables has been often overseen or done wrong (including in one of the first author's earlier papers, Turowski et al., 2007). So, one of the aims of this chapter is the clarification of this particular relationship and the provision of a sound physical basis to model it. We have added a few sentences at the beginning of the chapter to clarify this aim.

Q* is also not defined in your notation.

Is Q* on the x-axes in figure 4 supposed to be Q*_s with a bar over it? Maybe same
for Figure 5 and also in the caption of figure 4?
*Sorry, this was carried over from an early version of the manuscript. Of course we meant Q*_s.
We have corrected throughout.*

L 465 Where did 5.7/11.5 come from?
*These are dimensionless constants determined from experiment by Fernandez-Luque and van
Beek (1976). See equations 35/36. We provided a reference to these equations and the text has
been edited for clarification.*

Figure 10A – the x-axis is not formatted correctly.
*We do not see exactly the problem the reviewer tries to point out. However, we have revised all
figures to improve clarity and readability.*

Sentence starting on L 611: Does this mean that A* = 0 at all times?
*In the natural channel bed of the Erlenbach, yes. But this is irrelevant here, as we were trying to
say with the criticized sentence. We merely use the measured time series of bedload transport
and discharge to supply realistic input data. We have revised the introduction to the chapter to
make this clear.*

I guess this gets to a point that I didn't understand in the previous section - when you are talking
about the evolution of the mass on the bed through time, this could be under conditions of
complete cover the entire time. It might be worth stating this directly. Although I'm not sure that
I'm correct about this assumption. In Figure 12, assuming that A on the second from top plot is
actually A*, implies that there is not complete cover. I don't think I understood this section very
well.
*Yes, of course, the cover can be complete all the time. In the simulations we present here for the
Erlenbach, the final cover state depends strongly on the initial cover state. This indicates that
during the event, there was not enough time to reach a steady cover state. If we had started the
simulation with full cover, it would have stayed full for the entire event.
That said, the temporal evolution is also strongly dependent on how one calculates transport
capacity, which is a thorny issue for steep streams such as the Erlenbach. We have normalized
our transport capacity such that the ratio of supply to capacity is equal to one at the highest
discharge. Thus, during most of the event, relative sediment supply is smaller than one (in effect,
by choice). The point here was to show the cover evolution for realistic boundary conditions as
an illustrative example, rather than trying to predict bed cover throughout an event for the
Erlenbach. We have rephrased to improve clarity.*

Also in this section, in a single flood event I'm having a hard time understanding what
the meaning of p, or the period is. The previous section discussed the period in terms
of a sine wave in sediment supply. But is that the case for this flood?
*This is meant to be the same as the period discussed earlier. In essence, we assumed that at
each time step, a new sinusoidal perturbation with a fixed amplitude commences. From the data
we can estimate the local gradient in the variables and use this to calculate an 'effective' period.
We agree that we had not done a good job in explaining this and have tried to improve.*

Sentence starting on L 669: This is great. Is this shown explicitly and I didn't catch it?
If not, can you spell this out more directly?
*As is explained in the paragraphs preceding this statement, it is currently not that easy to
provide direct comparisons with data or calibrate the P-function directly, mainly because the
necessary parameters have not been reported in the experimental studies. However, we have
directly demonstrated that the formulation is flexible enough to yield a wide range of cover*

*functions (see Figs. 4 and 5 for examples). These do encompass (most of) the relationships observed in laboratory and numerical experiments. We do not think that we can meaningfully go any further at the moment, but have provided a reference to the figures.*

L 678: This dynamics cover ... typo I think.
*Typo; corrected.*

Sentence starting on Line 735 has a typo.
*Indeed; corrected.*

Section 4.3: The comparison with Philips and Jerolmack is a bit confusing to me. You state that in contrast to their findings, your findings suggest that bed cover is adjusted. But you didn't actually have channel morphology as a free variable. So I'm not sure how this study can contrast that one. I don't remember exactly, but I don't think they talked about sediment cover. So is this a fair way to compare the two studies?
I like how your conclusions stated this issue - both cover and channel morphology evolve. This makes sense to me, but the discussion in section 4.3 did not.
*Here, a main point is that bedrock rivers can adjust cover to achieve grade. And this can be done much more rapidly than adjusting channel morphology. Philips and Jerolmack missed this mechanism – they argue that in bedrock channels, morphological adjustment needs to be quick, because they observe graded channels. But, as stated, the river has other options to achieve grade – by adjusting bed cover. We have rewritten the entire paragraph to clarify.*

Reviewer #2

In this manuscript, the authors presented a probabilistic framework for predicting partial cover in mixed bedrock-alluvial channels, which they used to explore how probability of sediment deposition, relative sediment supply, and particle speed interact. It represents a next step in the progress that has been made over the past several years in modeling areal fraction of sediment cover. Overall, I found this to be a good paper, with sound methods and interesting results.
*We thank the reviewer for this assessment and the comments. We hope that in our revisions we can satisfactorily answer all queries.*

1. My main concern with the manuscript is the equations for entrainment rate and deposition rate. Although the authors explain that eq. 20 approaches $E_{max}^*$ as $M_s^*$ goes to infinity, does the same apply to eq. 21? I think $M_m^*$ cannot be infinity because it is limited by the capacity value $M_0^*$. If so, when $M_s^*$ is very large, it is impossible to balance D with E.
*$M_m$ is related to particle speed and upstream sediment supply via equation (24). Since both of these other variables are treated as input parameters, $M_m$ is fixed by them (at steady state). $M_m$ can only become infinite in the non-physical cases of zero particle speed but finite transport rate, or infinite supply. The deposition rate is then set by eq. (21). If $M_m$ is large (i.e., enough sediment is available), deposition is limited by upstream sediment supply. In essence, eq. (21) states that when $M_m$ is small, the amount that can be deposited is limited by $M_m$ (if there is only one mobile particle available, then a maximum of one can be deposited), and if $M_m$ is large, deposition is limited by sediment supply. This explanation has been added to the text.*

2. The authors assume that increasing sediment deposition decreases local shear stress and increases the critical entrainment shear stress for grains (Line 186-192 and 242-245). I think this assumption is limited to the case of smooth bedrock. Sediment deposition does not necessarily decrease the flow velocity. In rough bedrock, increasing sediment deposition increases local shear stress and decreases the critical shear stress for grains.

*This is a good point that we have taken up in the discussion. Indeed, we were thinking of smooth rather than rough bedrock. We have added 'smooth' at the appropriate places to qualify. In addition, feedbacks between cover and roughness have been discussed in some depth in the introduction.*

3. In section 4.2, although the authors explain the differences from the model presented by Nelson and Seminara (2011, 2012); the model presented in this paper has more similarities to the model presented by Inoue et al. (2014). In the mentioned paper, they have distinguished between mobile and stationary sediment, and have not assumed a direct correspondence between sediment concentration and degree of cover, which is different from Nelson and Seminara (2011, 2012). I think the main difference is in the sediment continuity equation including entrainment rate and deposition rate. This equation seems very useful. I encourage the authors to explain the differences from Inoue et al. (2014) and the advantages of this sediment continuity equation.

*We have included an additional paragraph discussing this model.*

Additional comments by line number below:
Line 140: There are situations when sediment does not accumulate even if the exposed part is zero. For example, runaway alluviation in Chatanantavet and Paker (2008).

*We are aware of this mechanism and mention it several times in the manuscript. In principle, the mechanism can be modelled in the framework, but entrainment and deposition rate for this case need to be dependent on the state of cover. As a purely descriptive tool, the P-function will still work (there is an example of run-away alluviation in the model data). There could potentially also be hysteresis in cover, which could be modelled by separate P-functions for entrainment and deposition. This is, however, beyond the scope of the paper, in particular as there are currently no data available to constrain the P-function for laboratory experiments or natural channels.*
*We do not see why the reviewer made this statement in relation to line 140 of the manuscript. All statements there are general and correct.*

Line 242-245: It is good to describe a physical reason for smoothing.
Smoothing is applied to prevent the formation of unrealistic piles of grains in one cell when there are far fewer grains in adjacent cells. The text was edited to explain.

Line 345: Can $(1-e^{-Ms})qt^*$ be converted to $(1-A)qt^*$? If so, the entrainment rate is proportional to the areal fraction of sediment cover.
*Only for the assumption $P=A^*$. We work with this assumption a lot in the later interpretation of the equations and in examples, but at this point, P is left general. But this is a nice way of looking at it.*

Figure 4 and Figure 5: $Q^* = Qs^*$?
*Apologies, this was a mistake.*

Figure 7: What is arbitrary unit? Is the sediment supply rate specified?
*One uses arbitrary units when the absolute scale of the relation is irrelevant for the argument.*

Figure 8: Please explain the distance from upstream end to downstream end, transport capacity, bed slope and grain size.

*All the variables mentioned by the reviewer are not relevant – what is relevant is not the absolute transport capacity, but the relative sediment supply (supply normalized by transport capacity). We have worked here in the non-dimensional framework specified in 3.1. We have revised the caption for Figure 8, including references to the relevant equation and the one variable that we had missed, particle speed. All relevant information is now contained in the caption.*

Figure10: Which equations are used to calculate 99% response time?

*We used numerical solutions to obtain these data. The corresponding equations in the paper are (3), (22), (23) and (24). The text has been added to clarify.*

[revised manuscript text omitted]

---

## Author Response (AR2)

**Rebuttal and track-changes manuscript**
A probabilistic framework for the cover effect in bedrock erosion
Submitted to Earth Surface Dynamics
Jens M. Turowski & Rebecca Hodge

Comments to the Author:
First, I would like to thank the reviewers for many constructive comments, and the authors for preparing and submitting a revised version of the manuscript. I think that the manuscript has been improved and it is on the right track.
However, that said, I also think that there are places where improvements and clarifications are needed before publication.
*We thank the editor for his comments. We have tried to further improve the clarity as much as possible.*

I recommend that the authors at this stage continue revision of the manuscript based on my comments to the revised version of the manuscript. My comments are listed below. In this process, the authors should use any potential there is to shorten the text, in particularly section 3, which I found difficult to read.
*We do not think that shortening section 3 would aid the clarity of the paper, and therefore have not done so. We think this section contains the most interesting and useful results of the paper. We have, however, reworked the text and provided an introduction to what we intend to do there and where the significance lies. See below for more details.*

General comments:

As a first general comment, I must say that I agree with reviewer 1 in that the detailed level of the math hampers the reading of the paper, and I do not think the authors have done enough to meet the concerns of the reviewer here. In my view, the problem is not the equations themselves, nor is it the number of equations. However, the lengthy deviations are in places poorly motivated, and I had several times to read ahead in the paper to understand where the math was going.
*We have added explanatory statements of what we are trying to do to the start of each section to prepare the reader for what is to come.*

It is particularly a problem in section 3, which I think could be significantly shortened, for example by moving the non-dimensional analysis in section 3.1 or the steady-state analysis in section 3.2 to the appendix. In stead, I suggest that more use could be made of the numerical solution to eqn. 13 (Fig. 8). Does the numerical solution coupled with intuition not illustrate most points (e.g. line 615, 780-782) made about steady-states and the transient transitions between them? I do not question the validity of the analytical solutions, but I wonder if they represent the best way to communicate your points.
*We disagree strongly here. We believe the analytic solutions of the steady state cover and of the time scale are the core results of the paper and provide the most useful outcome and have emphasized this point in the paper. This comes partly from the history of the subject; the relation between the exposed fraction and transport stage, as given in eq. 1 and 2, and later in eq. 27 for our model is the common form of the cover function in the literature. We believe it is essential to keep the derivation and discussion of this equation in the main text, as it provides the link to previously published material. Here, we derive an analytical, physically-based (and mathematically correct) expression of this equation for the first time and, again, this is a core result of the paper.*

*In our mind, the numerical solution is uninteresting beyond illustrating a simple example and*
*preparing the following considerations, as it is always tied to specific cases, while the analytical*
*solutions give some general insight into the system behavior. The system time scale in particular*
*allows estimates of whether in a given situation, a dynamic solution is necessary or whether cover*
*can be calculated with steady state equations. The chapter therefore provides a natural continuation*
*of the preceding section. We have reworked the introduction to the section to make clearer the*
*intentions and the significance of the derivations. Again, this is a core result of the paper – dynamic*
*modelling was so far not possible, and only the steady state equations (mostly eq. 1) have been used,*
*although there were hints from field, laboratory and numerical work that these are insufficient and*
*the sediment dynamics need to be taken into account.*
In any case, you need to motivate the analytical solutions more and carefully explain the assumptions
made.
*At the start of the different sections, we have added motivations for the developments that follow.*
For example, what is the implication of assuming qs=0 in line 503? Under which conditions can this
be assumed, which limitations does the assumption introduce, and what new insights can expect
from studying this situation in particular.
*Here a full analytical solution can be obtained. As it has already been said the past versions, the*
*situation is not common in nature, but provides a simple laboratory test for the theory. This can be*
*used for example to constrain the P-function.*
Also, like reviewer 1, I was confused about Eqn. 13. At first glance it looks like a standard continuity
equation, but with double treatment of deposition/entrainment. I fully understand the reviewer's
comment, and it was only after reading your answer to the reviewer's comment that I understood
that this is the balance of only the mobile mass. When introducing the equation in line 314 you write:
"…we use the total sediment mass on the bed as a variable". I recommend rephrasing this and
explain carefully how the two reservoirs interact. I also recommend including a schematic figure as
you suggest, and to move Eqn. 16 up next to Eqn. 13 so that both balance equations for the two
sediment reservoirs are presented together. It is important here to carefully present the model to a
wide group of readers, instead of relying on the readers' knowledge of the Exner equation (which I
did not know, and would not have remembered if I did).
*We have reworked the introduction to the chapter and tried to explain the framework better. In*
*addition, we have included a cartoon and the mass balance difference equation. We have also*
*reorganized the equations, as suggested.*
Section 2 presents the probabilistic framework, and this section already reads well. However, there
are some issues with the implied relation between Ms and Qs. This relation is the main subject of
section 3, but it seems here that you are assuming something that the readers are not yet shown. For
example in line 155, you link eq. 6 directly to eq. 1 even though one is using Qs and the other is using
Ms. I failed the see why these two linear equations are identical. At least some clarification is
needed.
*We agree that we have not been careful in making connections here and have removed them.*
Like for section 3, I suggest that you consider introducing a simple schematic figure also in section 2,
perhaps of an evolving bed-cover in a channel. Such a figure could be used to illustrate the meaning
of A*, Ms*, P etc. It would also nicely supplement the many curves presented in the other figures.
*We added a cartoon. Hopefully this makes things clearer.*

As a final general comment, there is a tendency in the manuscript to use quite subjective, and at the
same time imprecise, descriptions of previous work. Some examples:
Line 720: "Despite short-comings, Aubert et at. (2015) presented…" What shortcoming? You need to
either skip the reference to shortcomings or carefully describe them.
Line 749: "While the model is interesting and provides…" What is interesting about it? Be more
specific please, or skip the descriptive wording.
Line 771: "…quickly in response to changing boundary conditions, a somewhat counter-intuitive
notion for slowly-eroding channels" – If this is counter-intuitive or not must depend on the precise
meaning of "quickly" and "slowly-eroding". Again be more specific or skip it.
*We have removed or rewritten these parts.*

More specific comments:

line 24: "they" -> channels
*Changed to passive mode, 'such that the supplied sediment load can just be transported'.*

line 79: "tended" – tend
*Changed.*

lines 99-101: consider rephrasing this sentence; I'm not sure that I understand it.
*Sentence has been rephrased; now it reads 'Using the CA model of Hodge and Hoey (2012), Hodge (in*
*press) found that, when sediment supply was very variable (alternating large pulses with no sediment*
*supply), the amount of sediment cover was primarily determined by the recent supply history, rather*
*than by the relationships identified under constant sediment supply.'*

lines 124-128: You could move some of the information given in lines 678-698 up here to better
motivate why influence of other parameters such as bed topography and roughness is ignored.
*To us it seems that this information only makes sense after one has been exposed to the dynamic*
*model developed in section 3. Instead, we argue that dealing with A\* only is sufficiently flexible, since*
*in the end all the other influences affect the development of A\*.*

line 162: move ",k," up to after "…value"; it will make k more visible.
*Changed.*

eq. 9: in eq. 2 you use exp here e – needs to be consistent.
*Changed to the exponential notation here. Although, both notations are standard and common, and*
*they are defined in the text.*

line 245: "though" -> "through"?
*Corrected.*

line 254: What is the model step length? Also is the probability of deposition always 1? If so, please
write so explicitly.
*Text updated to clarify this; now reads 'In each time step random numbers and the probabilities are*
*used to select the grains that are entrained, which are then moved a step length of ten cells*
*downstream and deposited. Model results are insensitive to the step length.'*

line 270: "… given out by the model…" please rephrase.
*Rephrased; now 'Cover bed fraction and total mass on the bed produced by the model were converted*
*using eq. (3) into the new probabilistic framework (Fig. 3).'*

line 271: Is the CA model not also a probabilistic model? Its control parameters seem to be
likelihoods of entrainment, whereas the "probabilistic framework" presented has likelihoods of
deposition. But still, in essence both are probabilistic, right?
*Yes, the CA model is also probabilistic, and this is explained in the earlier section. We do not think this*
*needs to be further emphasized here.*

Fig. 3: The comments above also makes me think that there should be more direct relationships
between P and A within your framework, and then pi and pc from the CA model?
*Clearly there is a relation between cover and pi and pc, since the latter two variables determine the*
*evolution and steady state value of the former. However, it is not possible to derive these relations*
*analytically. The 'empirical' connections are depicted in Fig. 3.*

Line 298: The relation between Qs* and Ms is not clear from eqn. 3.
*We have removed this reference.*

Line 299: what do you mean by "muddled"?
*We removed this sentence.*

Line 299: what is incorrect?
*We removed this sentence.*

line 300: "bases" -> "basis"
*Corrected.*

line 312: skip "of course"
*Deleted.*

line 325: while non-dimensional variables are certainly useful, they also introduce complexity. I did
not understand why they were necessary here. Please motivate the non-dimensional analysis better.
Give the reader some clear hints as to where the analysis is going (and consider moving it to
appendix).
*We do not think anything is gained by moving the dimensionless variables to the appendix. When the*
*equations are written with them, they need to be introduced in the main text and defined anyway,*
*and there is little more than that done in the text at the moment.*

line 335: "The rate of change of the stationary sediment mass…" seems a bit counter-intuitive text-
wise. To me "stationary" means "not moving". Could you replace "stationary sediment" with
"deposited sediment"? You would have to change this throughout the text though – but the reading
would be better I think.
*We disagree. The term 'deposited' makes a statement about the history of the sediment and we think*
*that 'stationary' is both clearer and more precise. The erosion-deposition framework that we exploit*
*here is well established in the theoretical fluvial geomorphology. A stationary particle becomes*
*mobile by entrainment and a mobile particle becomes stationary by deposition. We do not think that*

*the use of language as we have used it would be confusing to any fluvial geomorphologist and have*
*left the terminology as is.*
line 497: "the equations" – which ones?
*Added equation numbers.*
line 503: Please remind the reader about the meaning of $q_s*$ and $q_t*$. What is happening when $q_s*=0$
and what is the relevance of this case?
*We provided an introduction to this chapter, explaining the motivation and significance and moved*
*the motivation for the specific case to the beginning of the section.*
line 663: "1000s" -> "1000 s"
*Changed.*
line 717: "depended" -> "depends" (if Aubert et al.'s results are still valid then consider using present
tense in such places.
*Changed.*
line 730-735: I could not follow these arguments about grain size and grid size.
*If the grid size is equal to the grain size, each deposited grain covers an entire grid square. If the*
*grains are smaller than the grid size, then the distribution of grains within each node determine the*
*overall cover.*
*We have added explanation: 'Further, Nelson and Seminara (2011, 2012) assumed a direct*
*correspondence between sediment concentration and degree of cover, which is equivalent to the*
*linear cover function (eq. 6). In this case, it is assumed that grains are always deposited on uncovered*
*bed and the different possible distributions of particles within a grid node are not taken into account.*
*Practically, this implies that the grid size needs to be of the order of the grain size, because, strictly,*
*the assumption is only valid if a single grain can cover an entire grid node.'*
line 751-753: I'm still not sure how this would work. Can you be more specific in how bed roughness
could be implemented in the framework?
*The P-function can be dependent on roughness (see also the statements following its introduction in*
*eq. 3). We added a statement to the sentence:*
*'In principle, the probabilistic framework presented here should be able to deal with macro-rough*
*beds, by making the P-function (eq. 3) explicitly dependent on roughness, and thus allows a more*
*general treatment of the problem of bed cover.'*
line 762: Also here you could be more precise. It is a general point of this manuscript that the
probabilistic framework can me integrated with many types of data, but I'm still not sure how. More
guidance would be useful.
*If data on stationary sediment mass and cover is available for various steady cover states, one can*
*just take the derivative according to eq. 3 and derive P from that. It does not matter whether the data*
*were derived from the field, or from laboratory or numerical experiments. The example in section 2.2*
*was put in to illustrate this. The process of conversion was explained in the first sentences of the last*
*paragraph of this section (starting line 270 in the old version of the manuscript and line 275 in the*
*new version).*
*Nevertheless, we agree that the statement sits somewhat oddly at the point of the discussion referred*
*to here and have removed it.*

line 771: "Cln" – "In"
*Corrected.*

[revised manuscript text omitted]